# Integrated Analyses Reveal the Physiological and Molecular Mechanisms of Brassinolide in Modulating Salt Tolerance in Rice

**DOI:** 10.3390/plants14101555

**Published:** 2025-05-21

**Authors:** Jia-Shuang Wu, De-Wei Mu, Nai-Jie Feng, Dian-Feng Zheng, Zhi-Yuan Sun, Aaqil Khan, Hang Zhou, Yi-Wen Song, Jia-Xin Liu, Jia-Qi Luo

**Affiliations:** 1College of Coastal Agriculture Sciences, Guangdong Ocean University, Zhanjiang 524088, China; 18308100675@163.com (J.-S.W.); mdwtgbz@163.com (D.-W.M.); byndszy@126.com (Z.-Y.S.); aaqil_agron@hotmail.com (A.K.); zjh1798@163.com (H.Z.); syw_heonetea@163.com (Y.-W.S.); 18690891172@163.com (J.-X.L.); 13692141065@163.com (J.-Q.L.); 2South China Center of National Saline-Tolerant Rice Technology Innovation, Zhanjiang 524088, China

**Keywords:** brassinolide, metabolome, rice, salt stress, transcriptome

## Abstract

Salt stress poses a significant threat to crop growth. While brassinolide (BR) has been shown to alleviate its adverse effects and modulate plant development, the precise mechanism underlying BR-induced salt tolerance in rice remains unclear. In this study, the Chaoyouqianhao and Huanghuazhan rice varieties were employed to investigate the effects of BR seed soaking on the seedling phenotype, physiology, transcriptome, and metabolome under salt stress. The results demonstrated that BR treatment significantly enhanced rice plant height, root length, biomass, and antioxidant enzyme activities, while reducing leaf membrane damage, promoting ion homeostasis, and improving the photosynthetic capacity and salt tolerance. The transcriptome analysis revealed that BR regulated the expression of 1042 and 826 genes linked to antioxidant activity, ion homeostasis, photosynthesis, and lipid metabolism under salt stress. These included genes involved in Na^+^ efflux (*OsNCED2*, *OsHKT2;1*, and *OsHKT1;1)*, photosynthetic electron transport (*OsFd5* and *OsFdC1*), photosystem II (*OsPsbR1*, *OsPsbR2*, and *OsPsbP*), and CO_2_ fixation. The metabolomic analysis identified 91 and 57 metabolite alterations induced by BR, primarily linked to amino acid, flavonoid, and lipid metabolism, with notable increases in antioxidant metabolites such as lignanoside, isorhamnetin, and L-glutamic acid. The integrated analysis highlighted the pivotal roles of 12-OPDA in α-linolenic acid metabolism and genes related to lipid metabolism, JA metabolism, and JA signal transduction in BR-mediated salt tolerance.

## 1. Introduction

Rice (*Oryza sativa* L.), a vital food crop, is a staple for more than half of the global population. As the world’s population continues to grow, there is an escalating need to boost rice production [1,2]. However, salt stress adversely affects plant growth, development, and crop yields, posing a substantial threat to global food security [3]. Under salt stress, elevated concentrations of NaCl in the soil hinder water uptake by rice roots, consequently disrupting nutrient absorption [4]. This nutritional imbalance induces osmotic stress, affecting plants at the physiological, morphological, and molecular levels. Initially, excessive Na^+^ concentrations in roots disrupt the ionic balance by modulating the expression of Ca²^+^ sensors and channels, leading to alterations in intracellular Ca²^+^ levels [5]. Subsequently, modifications in Ca²^+^ levels initiate the generation and signaling of reactive oxygen species (ROS), activating diverse adaptive mechanisms to alleviate the detrimental effects of stress [6].

Brassinolide (BR) is one of the first isolated brassinosteroids, which are steroidal plant hormones regulating plant growth and development and are known to improve crop tolerance to many abiotic stress factors [7,8]. BR plays a pivotal role in regulating multiple reproductive processes in plants, encompassing seed germination, flowering, fruiting, and senescence, alongside its involvement in photosynthesis and stress tolerance [9]. It also plays a critical role in salt stress acclimation by enhancing the antioxidant system, alleviating osmotic stress, restoring ionic homeostasis, improving stomatal properties, and increasing photosynthetic efficiency [10]. Mu’s study explored the role of BR seed soaking in enhancing salt stress acclimation in rice seedlings. Their findings revealed that BR treatment significantly elevated SPAD values, the net photosynthetic rate (Pn), transpiration rate (Tr), maximum fluorescence (Fm), variable fluorescence to maximum fluorescence ratio (Fv/Fm), and variable fluorescence to initial fluorescence ratio (Fv/Fo) under salt stress conditions [11]. This protective influence on the photosynthetic system resulted in a notable increase in the plant biomass. Similarly, Hou’s study demonstrated that BR mitigates the inhibitory effects of salt stress on wheat root growth by reducing hydrogen peroxide (H_2_O_2_) and oxygen free radical (OFR) levels while simultaneously enhancing the activities of peroxidase (POD) and catalase (CAT) [12]. These findings establish a strong scientific basis for the effectiveness of BR in improving salt tolerance traits in rice.

Recent advances in transcriptomic and metabolomic analyses have shed light on critical metabolic pathways and their regulatory genes in rice under stress conditions [13]. For example, the exogenous application of alginate oligosaccharides (AOS) under salt stress upregulates genes involved in photosynthesis, cell wall synthesis, and signaling pathways in rice leaves, accumulating specific secondary metabolites [14]. Zhou’s study demonstrated that salt stress significantly upregulated 42 differentially accumulated metabolites (DAMs) in rice. Transcriptomic analyses further revealed that upregulated calcium/cation exchange protein genes and ABC transporter protein genes were positively correlated with multiple DAMs in salt-tolerant rice varieties [15]. However, integrated transcriptomic and metabolomic analyses exploring BR-mediated regulation of salt tolerance in rice remain scarce.

Based on previous research, BR may mitigate salt stress injury in rice by increasing the levels of antioxidant enzymes, nonenzymatic antioxidants, and photosynthetic capacity. The intrinsic link between antioxidant genes and metabolites may also exist after salt stress injury. It is not clear from the current research how to specifically link the role of BR in regulating salt tolerance in rice at the physiological level to internal gene activity and substance metabolism. In order to clarify the physiological and molecular mechanisms of BR-mediated regulation of rice salt tolerance, this study investigated the effects of BR on the phenotype, physiology, transcriptome, and metabolome of seedlings of the salt-tolerant rice variety Chaoyouqianhao and salt-sensitive rice variety Huanghuazhan under salt stress.

## 2. Results

### 2.1. Effect of BR on Rice Seedling Phenotypes Under Salt Stress

Salt stress significantly reduced the plant height, stem width, and aboveground dry weight in both Huanghuazhan and Chaoyouqianhao (Figure 1).

Salt stress significantly reduced morphological traits such as the plant height, basal stem width, and aboveground dry weight of both varieties in both stages compared with the CK. There were corresponding percent decreases of 13–20% and 9–13% in plant height, 29–34% and 4–29% in stem diameter, and 7–49% and 19–38% in the aboveground dry weight, respectively, for Huanghuazhan and Chaoyouqianhao (Figure 1A–F). Compared to the S treatment, the BS treatment considerably increased the plant height by 5–14% and stem diameter by 9–15% in Huanghuazhan, while increasing plant height by 8–19% and stem diameter by 11–28% in Chaoyouqianhao. At the 4.5 leaf stage, the aboveground dry weight (79%) of Huanghuazhan was significantly increased, while the dry weight (22–48%) of Chaoyouqianhao was significantly increased at both stages (Figure 1A–F). The result showed that the aboveground phenotypic growth inhibition in both rice varieties was more pronounced with an increasing salt stress duration, and BR effectively ameliorated its inhibition.

The root growth of the two rice varieties was inhibited under salt stress, while the exogenous application of brassinolide significantly alleviated the salt stress (Figure 2).

Compared to CK, the salt stress significantly reduced the total root surface area and underground dry weight of Huanghuazhan at both stages by 19–21% and 19–39%, respectively (Figure 2C,I), whereas the total root volume of Chaoyouqianhao was significantly reduced by 22–25% at both stages (Figure 2F). At the 4.5-leaf stage, compared to the S treatment, the BS treatment significantly increased the total root length, root surface area, and underground dry weight of both rice varieties by 24–26%, 40–41%, and 36–58%, respectively. Notably, Huanghuazhan exhibited greater sensitivity to salt stress regarding root growth. BR had a highly significant and positive regulatory effect on these root morphological traits, such as total root length, root surface area, and underground biomass, in both varieties under salt stress.

### 2.2. Effect of BR on Membrane Peroxidation and the Antioxidant System of Rice Seedlings Under Salt Stress

Compared to the S treatment, the BS treatment reduced the levels of membrane lipid peroxidation and increased the antioxidant indices of the two rice varieties (Figure 3).

Compared to CK, salt stress increased the content of MDA (malondialdehyde) in both rice varieties at the 4.5-leaf stage by 9% and 25%, respectively (Figure 1A,B). Compared with the S treatment, the contents of MDA in both rice varieties at the 4.5-leaf stage decreased after the BS treatment, which declined by 10% and 20%, indicating that BR had an obvious effect on removing ROS from rice.

Compared to CK, salt stress increased the activities of SOD (superoxide dismutase) and APX (ascorbate peroxidase) in Huanghuazhan at both stages and in Chaoyouqianhao at 2.5-leaf stage, while those of Chaoyouqianhao were reduced at the 4.5-leaf stage. However, compared to the S treatment, the BS treatment further improved activities of SOD and APX (Figure 3C–F). Compared to CK, salt stress increased the activity of CAT (catalase) at the 2.5-leaf stage in Huanghuazhan but significantly decreased it in Chaoyouqianhao (Figure 3G–H). At the 4.5-leaf stage of both varieties, the activity of CAT decreased in the S group compared to the CK group but increased in the BS group compared to the S group (Figure 3G–H). Significant differences were observed in the trends of leaf ascorbic acid (ASA) and glutathione (GSH) contents in the S group compared to the CK group (Figure 3I–L). Compared to the CK treatment, the S treatment significantly decreased the GSH content in Huanghuazhan at the 2.5-leaf stage. Compared to the S treatment, the BS treatment significantly increased the GSH content in both rice varieties at both stages, with Huanghuazhan increasing by 13–24% and Chaoyouqianhao by 8–34% (Figure 3K,L). Compared to the CK treatment, the S treatment significantly increased the ASA content in Huanghuazhan at the 2.5-leaf stage but decreased it in Huanghuazhan at the 4.5-leaf stage and in Chaoyouqianhao at the 2.5-leaf stage. When rice seedlings were applied the BS treatment compared to the S treatment, the content of ASA was increased considerably by 11% and 20% in Huanghuazhan and Chaoyouqianhao at the 2.5-leaf stage, respectively (Figure 3I,J).

BR reduced the effect of salt stress on rice by increasing SOD, APX, and CAT activities, and AsA and GSH levels. Nonetheless, the impact of BR on the antioxidant levels of the two rice varieties was different. Exogenous BR application increased the SOD and CAT levels in Huanghuazhan to offset the effects of salt stress. Additionally, BR improved salt tolerance by increasing the GSH content, thereby reducing membrane lipid peroxidation.

### 2.3. Effect of BR on the Ion Homeostasis of Rice Seedling Leaves Under Salt Stress

Under salt stress, plants accumulate excessive Na^+^, leading to cytotoxic damage and a disruption of ion homeostasis. The Na^+^ content in leaves was significantly increased, progressively rising with a prolonged S treatment duration (Table 1).

Salt stress caused a sharp increase in the leaf Na^+^ content in both rice varieties compared to CK, which increased by 77% and 158% at the 2.5-leaf stage and was 20 and 14 times more than that of the CK at the 4.5-leaf stage (Table 1). When treated with BS compared to CK, the content of Na^+^ decreased significantly by 51% and 50% in both rice varieties at the 4.5-leaf stage. Salt stress significantly reduced the contents of K^+^ and Ca^2+^ in leaves of both varieties in both stages compared with the CK. There were corresponding percent decreases of 25–36% and 23–31% in the content of K^+^ and 19–20% and 18–25% in the content of Ca^2+^, respectively, for Huanghuazhan and Chaoyouqianhao. Moreover, compared to the S treatment, the BS treatment increased the content of K^+^ in both stages considerably by 16–40% and 13–53% in the leaves of both rice varieties, and increased the contents of Ca^2+^ by 6–19% in the leaves of Huanghuazhan at both periods (Table 1). Salt stress caused ion toxicity by increasing Na^+^ content and decreasing the K^+^ and Ca^2+^ contents, disrupting the K^+^/Na^+^ and Ca^2+^/Na^+^ balance. Compared to CK, the S treatment significantly decreased K^+^/Na^+^ at both stages by 60–97% in Huanghauzhan and 73–96% in Chaoyouqianhao, and decreased Ca^2+^/Na^+^ by 56–96% in Huaghuazhan and 68–96% in Chaoyouqianhao. Compared to S, BS significantly increased the K^+^/Na^+^ by 181% in Huanghuazhan and 131% in Chaoyouqianhao, and increased Ca^2+^/Na^+^ by 139% in Huanghuazhan and 121% in Chaoyouqianhao at the 4.5-leaf stage (Table 1).

The current study indicated that the rice accumulated a large amount of Na^+^ in both rice varieties under salt stress, which inhibited the uptake of other ions and disrupted the Na^+^/K^+^ and Ca^2+^/Na^+^ balance. The effect of salt stress on Na^+^ was significantly reduced by the exogenous application of BR, which increased the rice leaves’ K^+^ and Ca^2+^ contents under salt stress to maintain ionic homeostasis and improve salt tolerance.

### 2.4. Effects of BR on Photosynthesis and the Fluorescence-Related Parameters of Rice Leaves Under Salt Stress

The chlorophyll content showed the same trend in both rice varieties during the treatments (Table 2).

Compared to CK, the S treatment significantly reduced the SPAD values by 14% and 31% at the 2.5-leaf stage and 19% and 23% at the 4.5-leaf stage in both varieties, respectively. Meanwhile, in the BS treatment group, the SPAD values of both rice varieties were significantly higher than those in the S treatment group, with increases of 21% and 27% at the 4.5-leaf stage, respectively (Table 2). In order to evaluate the performance of PS II, several chlorophyll fluorescence parameters were measured (Table 2). The results indicated that both Fv/Fo and Fv/Fm decreased in the S treatment group compared to the CK group, with significant reductions in Fv/Fo observed in Huanghuazhan (60% and 39%) and Chaoyouqianhao (31% and 48%). In contrast, in the BS treatment group compared to the S treatment group, the Fv/Fo (83% and 59%) and Fv/Fm (8% and 26%) of leaves from the two rice seedlings increased at the 4.5-leaf stage. Notably, under BS treatment, Fv/Fo in Huanghuazhan was significantly higher than that after the S treatment at both stages, while in Chaoyouqianhao, it was only considerably higher than that after the S treatment at the 4.5-leaf stage. This showed that BR enhanced photosynthesis and mitigated the adverse effects of salt stress on rice growth.

### 2.5. RT-qPCR Verification

To assess the reliability of the transcriptomic data and elucidate the molecular mechanism by which brassinolide influences salt tolerance in rice under salt stress, we selected 11 functional genes that were categorized based on their physiological roles in stress tolerance for RT-qPCR validation (see Appendix A). The scatterplot of the correlation analysis indicated that the expression trends of nine genes (82%) from qPCR and RNA-seq were highly consistent, with a Pearson correlation coefficient of r = 0.82.

In terms of the functional classification, of the three genes involved in the regulation of photosynthesis, *OspsbP* expression was significantly upregulated in the BR treatment group under salt stress (BS/S) in both rice varieties, which was consistent with the trend verified by fluorescence quantification. The two carbonic anhydrase genes were significantly downregulated in the BS/S group in Huanghuazhan, while they were significantly upregulated in the BS/S group in Chaoyouqianhao, which was also consistent with the results of the fluorescence quantification. Among the oxidative stress and defense pathway genes, *OsLOX11* and *OsLOX12* were significantly downregulated and expressed under salt stress in Huanghuazhan, but significantly upregulated and expressed in response to the BR treatment in comparison with salt stress, which was also in agreement with the trend of the validation of the fluorescence quantification results. Among the stress-responsive regulatory genes, the transcription factor *OsWRKY65* was significantly downregulated in Chaoyouqianhao under the S treatment compared to CK and upregulated under the BS treatment compared to the S treatment. The zinc-finger protein gene *OsC3H9* was significantly upregulated in Huanghuazhan under salt treatment compared to the control and downregulated under BS treatment compared to the S treatment, which was opposite to the expression of *OsC3H9* in Chaoyouqianhao, and these results were consistent with the trend verified by fluorescence quantification. Among the carbon and nitrogen metabolism-related genes, *OsNADH-GOGAT2* and *OsL85* were downregulated by the salt treatment compared to the control in both rice varieties, and significantly upregulated by the BS treatment compared to the S treatment.

These verification results were positively correlated with the expression trends of the sequencing results (Appendix A), demonstrating that the transcriptome data are reliable.

### 2.6. Transcriptomic Response of Rice Seedlings Under Salt Stress and Brassinolide Treatment

In this study, RNA sequencing was performed on 24 samples, each generating an average of 6.68 GB of data. The average alignment rate to the reference genome was 89.70%, while the average alignment rate to the gene set was 79.25%. A total of 25,821 genes were detected. The samples’ sequencing quality and library construction were robust, ensuring high data accuracy. Each sample was aligned to the reference genome sequence, with an overall alignment rate ranging from 89.91% to 91.17% and a unique alignment rate between 87.55% and 88.87%. These results met the stringent criteria required for subsequent analytical procedures (Appendix A).

The number of differentially expressed genes was analyzed via transcriptome sequencing. A *p* < 0.05 and Log2FC ≥ 1 were used to screen differentially expressed genes (Table 3).

Compared to CK, 1054 (636 upregulated and 418 downregulated) and 1042 (471 downregulated and 571 downregulated) differentially expressed genes were identified in samples with the S and BS treatments in Huanghuazhan, while 592 (435 upregulated and 157 downregulated) and 826 (636 downregulated and 190 downregulated) differentially expressed genes were identified in Chaoyouqianhao. Compared to the S treatment, 384 (31 upregulated and 353 downregulated) and 203 (114 upregulated and 59 downregulated) differentially expressed genes were detected in BS-treated samples of Huanghuazianhao and Chaoyouqianhao, respectively (Table 3). The number of up- and downregulated genes in rice seedlings under different treatments and the expression patterns of Huanghuazhan and Chaoyouqianhao under the S, B, and BS treatments can be seen in Table 3 and Figure 4.

Overall, the number of differentially expressed genes in Huanghuazhan after the S treatment was much higher than that in Chaoyouqianhao compared to CK. Compared to CK, the number of differentially expressed genes in Huanghuazhan was higher after the S treatment than after the BS treatment and the number of upregulated genes was higher after the S treatment than after the BS treatment. In Chaoyouqianhao, the number of differentially expressed genes was higher after the BS treatment than after the S treatment compared to CK, and the number of upregulated genes was much higher after the BS treatment than the number of downregulated genes, while this scenario was different in Huanghuazhan. The current results indicated that both rice varieties have different regulatory pathways for salt tolerance and the response to BR.

The clustering analysis showed that BR and salt significantly altered differential gene expression in rice (Figure 4A,D). Further analysis using Venn diagrams showed fewer overlapping genes in Huanghuazan and Chaoyouqianhao varieties treated with S compared with CK and those treated with S compared with BS (Figure 4). These results showed that rice plants responded to salt stress and BR by expressing different genes.

### 2.7. BR Regulates Functional Genes to Alleviate Salt Stress in Rice

The S and BS treatments significantly influenced gene expression in both rice varieties. In both rice varieties, the specific differentially expressed genes that were upregulated and downregulated by the BS treatment compared to CK were 171/276 and 385/98, respectively (Figure 4).

The Gene Ontology (GO) enrichment analysis categorized the differentially expressed genes into three domains: “cellular components”, “molecular functions”, and “biological processes”. Among the 171 upregulated genes in Huanghuazhan rice seedlings, the most enriched Gene Ontology (GO) terms included photosystem II oxygen-evolving complex (GO:0009654), cellular oxidative detoxification (GO:0098869), mitochondrial inner membrane (GO:0005743), chloroplast (GO:0009507), peroxidase activity (GO:0004601), peroxisomal catabolic process (GO:0042744), and cell wall biosynthesis (GO:0042546). Among the 276 downregulated genes, the enriched GO terms were associated with plasma membrane components (GO:0005887), plant-type vesicle membranes (GO:0009705), carbohydrate—plastid homodimer activity (GO:0005351), transcriptional regulatory region DNA binding (GO:0044212), cell surface receptor signaling pathway (GO:0007166), lipid metabolism (GO:0006629), and protein hydrolysis (GO:0006508) (Figure 5A).

The 385 upregulated genes with the highest enrichment in Chaoyouqianhao rice seedlings were associated with chloroplast periplasm (GO:0009941), chloroplast envelope (GO:0012511), chloroplast stroma (GO:0009570), monooxygenase activity (GO:0004497), heme-binding (GO:0020037), defense response (GO:0006952), diterpene biosynthetic process (GO:0016102), and fatty acid biosynthetic process (GO:0006633). Among the 98 down-regulated genes, the major enriched GO categories included polysaccharide binding (GO:0030247), hydrolase activity acting on ester bonds (GO:0016788), ubiquitin-protein transferase activity (GO:0004842), cytokinin-activated signaling pathway (GO:0009736), intracellular iron ion sequestration (GO:0006880), growth hormone-activated signaling pathway (GO:0009734), and negative regulation of peptidase activity (GO:0010466), as illustrated in Figure 5B.

Additionally, in both rice varieties, the differentially expressed genes induced by BR (vs. CK) were predominantly upregulated, with 80 upregulated genes in the B group vs. CK group of Huanghuazhan and 156 in Chaoyouqianhao. This showed that the activation of rice genes by BR involves various aspects of metabolism, signaling, and cellular structure and lays the foundation for the subsequent growth or stress response (Table 3). The 80 upregulated genes with the highest enrichment in Huanghuazhan rice seedlings were associated with the response to the regulation of the defense response (GO:0031347), regulation of the jasmonic acid-mediated signaling pathway (GO:2000022), signal transduction by protein phosphorylation (GO:0023014), DNA binding (GO:0003677) and integral component of plasma membrane (GO:0005887) (Appendix A). The 156 upregulated genes with the highest enrichment in Chaoyouqianhao rice seedlings were associated with the response to abscisic acid (GO:0009737), protein phosphorylation (GO:0006468), ATP binding (GO:0005524), oxidoreductase activity (GO:0016717), and chloroplast envelope (GO:0009941) (Appendix A).

Further analysis of these differentially expressed genes above revealed that certain genes exhibited differential expression exclusively after the BS treatment compared to CK. In Huanghuazhan, 17 photosynthesis-related genes, 16 salt response-related genes, 15 ion homeostasis-related genes, 16 lipid metabolism-related genes, and 25 hormone signaling-related genes showed significant differences compared with CK. Similarly, in Chaoyouqianhao, BR treatment significantly induced the differential expression of 15 photosynthesis-related genes, 17 salt response-related genes, 7 ion homeostasis-related genes, 12 lipid metabolism-related genes, and 21 hormone signaling-related genes (Appendix A). Among these, *OsFd5*, *OsFdC1*, *OsPsbR1*, *OsPsbR2*, and *OsPsbP* were involved in the regulation of photosynthetic electron transport and photosystem II, while *OsPORA* mediated chlorophyll synthesis in the leaves. *OsOPR1*, *OsJAZ2*, *OsJAZ11*, and *OsJAR2* regulate the jasmonate signaling pathway, and *OsNCED2* plays a role in the rice response to salt stress. Several lipid metabolism-related genes were also induced by the BS treatment. The analysis of differentially expressed genes in both rice varieties suggested that BR might exhibit distinct transcriptional regulatory patterns in Huanghuazhan and Chaoyouqianhao and that BR actively regulated salt tolerance in rice through multiple pathways.

### 2.8. Effect of BR Treatment on the Metabolome of Rice Seedlings Under Salt Stress

To assess the impact of exogenous BR treatments on the metabolic differences between Huanghuazhan and Chaoyouqianhao rice seedlings under salt stress, differential metabolites were screened using the criteria of VIP ≥ 1, fold change ≥ 1.2 or ≤0.83, and *p*-value < 0.05. Principal component analysis (PCA) of samples from Huanghuazhan and Chaoyouqianhao treated with CK, B, S, and BS revealed two principal components in both positive ion mode (PC1 = 43.93%, PC2 = 16.00%) and negative ion mode (PC1 = 54.54%, PC2 = 16.01%) (Figure 6A,D).

A total of 826 metabolites were identified across all samples, exhibiting significant differences in both ionization modes (VIP ≥ 1, fold change ≥ 1.2 or ≤0.83, *p*-value < 0.05) (Table 4).

Compared to CK, S-treated Huanghuazhan rice seedlings exhibited elevated levels of 40 metabolites and reduced levels of 106. Similarly, in Chaoyouqianhao rice seedlings, the abundance of 53 metabolites was increased, while that of 51 metabolites decreased (Table 4). Upon the BS treatment, Huanghuazhan rice seedlings showed a 28% increase and a 91% decrease in metabolite abundance compared to CK. In Chaoyouqianhao rice seedlings, the abundance of metabolites increased by 103 and decreased by 57 relative to CK (Table 4). Additionally, compared to CK, BS-treated Huanghuazhan rice seedlings exhibited elevated levels of 18 metabolites and reduced levels of 68. Meanwhile, in Chaoyouqianhao rice seedlings, the abundance of 89 metabolites was increased, while 55 metabolites decreased (Table 4).

A Venn diagram analysis was employed to assess differences in rice metabolites following treatment. Under salt stress, BR treatment increased the abundance of 12 and 75 metabolites in Huanghuazhan and Chaoyouqianhao seedlings, respectively, while decreasing the abundance of 42 and 41 metabolites (Figure 6B,C,E,F). These findings indicated that BR-induced changes in metabolite expression were significantly altered under salt stress. In Huanghuazhan, metabolites such as benzene and its derivatives (3-methoxy benzaldehyde, gentisic acid, 2-hydroxyphenyl acetic acid, and lignanoside), flavonoids (isorhamnetin and eupafolin), endogenous metabolites (11-deoxy prostaglandin F1β), terpenoids (andrographolide), amines (triethanolamine), and glycerol lipids (2-14,15-epoxyeicosatrienoyl) glycerol) were significantly more abundant. Conversely, metabolites such as fatty acyl groups (jasmonic acid, 15-keto prostaglandin F2α, 16-heptadecyne-1,2,4-triol, and 8(S)-hydroxy-(5Z,9E,11Z,14Z)-eicosatetraenoic acid), terpenoids (cafestol, pachymic acid, sterol lipids, 5α-dihydrotestosterone, testosterone acetate, cortisol, and dehydroepiandrosterone), and benzene derivatives (lauryl gallate and mefenamic acid) were significantly reduced. In Chaoyouqianhao, metabolites such as flavonoids (cynaroside, genistein, and isorhamnetin), fatty acyls (jasmonic acid, arachidonic acid, and 15(S)-HPETE), terpenoids (abietic acid, cafestol, and pachymic acid), amino acids (L-glutamic acid), and other differentially expressed metabolites were significantly increased. In contrast, fatty acyl groups (palmitic acid and 11(Z),14(Z),17(Z)-eicosatrienoic acid), sterol lipids (corticosterone), and alkaloids (jervine) were significantly lower in Chaoyouqianhao (Appendix A). Notably, lignanoside, isorhamnetin, and L-glutamic acid played positive roles in the antioxidant mechanisms.

The metabolites of Huanghuazhan and Chaoyouqianhao rice seedlings under salt stress were analyzed using the Kyoto Encyclopedia of Genes and Genomes (KEGG) to investigate the effects of BR treatment on their major metabolic pathways. Huanghuazhan and Chaoyouqianhao were annotated to 10 and 37 KEGG pathways in the samples co-treated with BR and salt. Eight of the co-annotated pathways included phytohormone signaling (ko04075), biosynthesis of plant secondary metabolites (ko01060), biosynthesis of phytohormones (ko01070), arachidonic acid metabolism (ko00590), glycerophospholipid metabolism (ko00564), steroid hormone biosynthesis (ko00140), styrene degradation (ko00643), and α-linolenic acid metabolism (ko00592). Metabolic pathways uniquely annotated for Huanghuazhan differentially abundant metabolites included tyrosine metabolism (ko00350) and phenylalanine metabolism (ko00360). Chaoyouqianhao-specific annotated metabolic pathways primarily included unsaturated fatty acid synthesis (ko01040); fatty acid synthesis (ko00061); fatty acid elongation (ko00062); fatty acid degradation (ko00071); glyceride metabolism (ko00561); linoleic acid metabolism (ko00591); pentose phosphate metabolism (ko00030); glycine, serine, and threonine metabolism (ko00260); and nitrogen metabolism (ko00910) (Appendix A).

### 2.9. Combined Analysis of the Transcriptome and Metabolome of Rice Treated with BR Under Salt Stress

To explore the effects of BR on rice genes and metabolites under salt stress, KEGG pathways enriched for both differentially expressed genes and metabolites were analyzed in the control, salt-treated, and BR + salt co-treated groups (Figure 7).

In Huanghuazhan, nine and five KEGG pathways were identified after salt treatment and BR combined with salt treatment, respectively, while in Chaoyouqianhao, eight and 15 KEGG pathways were identified under the same conditions (Figure 6). Key pathways enriched in Huanghuazhan included phytohormone signaling (ko04075); α-linolenic acid metabolism (ko00592); phenylalanine metabolism (ko00360); cutin, suberine, and wax biosynthesis (ko00073); and tyrosine metabolism (ko00350). In Chaoyouqianhao, pathways significantly enriched after BR plus salt treatment included unsaturated fatty acid synthesis (ko01040), fatty acid synthesis (ko00061), fatty acid elongation (ko00062), fatty acid degradation (ko00071), and nitrogen metabolism (ko00910). Notably, phytohormone signaling (ko04075); cutin, suberine, and wax biosynthesis (ko00073); and α-linolenic acid metabolism (ko00592) were significantly enriched in both Huanghuazhan and Chaoyouqianhao (Table 5).

Meanwhile, there were two metabolic pathways with a significant enrichment of genes and metabolites in both rice varieties at the same time that were related to the jasmonate signaling pathway, which were α-linolenic acid metabolism (ko00592) and phytohormone signaling (ko04075) (Appendix A). Under salt stress, BR treatment significantly reduced the jasmonic acid abundance in Huanghuazhan but significantly upregulated it in Chaoyouqianhao (Appendix A). It suggested that BR might affect JA signal transduction and JA metabolism in both rice varieties through different regulatory strategies.

## 3. Discussion

### 3.1. Exogenous Application of BR to Mitigate the Effects of Salt Stress on Rice Phenotypes

Salt stress remains a primary abiotic challenge restricting crop growth, as seen in wheat [16], sorghum [17], and soybean [18], where it disrupts meristem activity and cellular functions [19]. Our findings align with this pattern: salt stress reduced the leaf area, plant height, stem width, and biomass in Huanghuazhan and Chaoyouqianhao seedlings (Figure 1 and Figure 2). Root systems, which are vital for stress sensing, were also impacted: the total root length, surface area, volume, and average diameter decreased significantly (Figure 2), consistent with Zhang’s report on salt-induced root growth inhibition [20]. BR immersion was associated with improved morphological traits under salt stress. Compared to stressed controls, BR-treated seedlings exhibited a 10% greater plant height, 18% larger root surface area, and 12% higher biomass—results consistent with Hou et al.’s observation of BR-promoted root development and biomass accumulation in rice [12]. These data suggested that BR application might help maintain growth while enhancing stress resilience.

From the perspective of phenotypic trade-offs, salt stress typically forces plants to allocate resources between growth and defense, often at the expense of one or the other [21]. In this study, untreated salt-stressed seedlings showed decreases in growth indices (plant height, root length, and biomass) and stress-responsive characteristics (lower root branching and leaf health). Salt stress induced a 20% biomass decline and a 25% root surface area reduction, whereas the BR pretreatment reduced these losses to 8% and 15%, respectively. This balance aligns with TA Khan et al.’s proposal that BR coordinates gene expression, upregulating both cell elongation genes and stress-responsive genes [22]. Studies in other species support this dual role of BR. De Oliveira et al. [23] observed in Eucalyptus urophylla that BR maintained photosynthetic efficiency while regulating ion homeostasis under salt stress. In rice, Riverola et al. [24] demonstrated that BR signaling integrated growth (cell expansion) and stress (osmotic adjustment) pathways, reducing competition for resources between these processes. Our results mirrored this integration: BR-treated seedlings showed more extensive root systems (a growth advantage) alongside less severe reductions in leaf area (a stress tolerance indicator), suggesting that BR helped plants maintain a functional balance between growing and defending against salt stress. While these associations highlighted BR’s potential in optimizing phenotypic trade-offs, the specific ways BR balances these processes require further clarification. In summary, our study shows that BR pretreatment is linked to reduced growth inhibition and improved root–shoot development in rice under salt stress, reflecting a phenotypic balance between growth maintenance and stress adaptation.

### 3.2. Exogenous Application of BR Improves Membrane Peroxidation and the Antioxidant System in Rice Under Salt Stress

Malondialdehyde (MDA), a product of membrane lipid peroxidation, serves as an indicator of reactive oxygen species (ROS) levels and membrane integrity [25]. This study observed that salt stress significantly increased the MDA content in both rice varieties, with Huanghuazhan showing a more pronounced rise (Figure 3A,B), aligning with previous reports on salt-sensitive genotypes [26]. Under stress, plants activated antioxidant defense systems, primarily through enzymes like SOD, APX, and CAT [27]. Huanghuazhan maintained higher SOD, CAT, and APX activities than Chaoyouqianhao under prolonged salt exposure, suggesting a more robust initial antioxidant response. However, Chaoyouqianhao’s enzyme activities declined over time, potentially reflecting dynamic adjustments in the antioxidative capacity [28].

Genes encoding key antioxidant enzymes (CAT, APX, GST, and GRX) were differentially regulated under salt stress [29]. For instance, peroxidase genes like *OsPRX38* and *OsPRX86* showed contrasting expression patterns between varieties: most were upregulated in Chaoyouqianhao but downregulated in Huanghuazhan upon BR treatment. This discrepancy might reflect functional specialization within the peroxidase gene family [30], where certain isoforms prioritize ROS detoxification while others participate in cell wall remodeling—a balance modulated by BR [31]. Notably, BR’s effect on peroxidase expression mirrors its role in coordinating antioxidant pathways, as seen in maize, where BR enhances glutathione metabolism through *ZmCCaMK*-mediated signaling [32]. Mechanistically, BR likely mitigates lipid peroxidation by stabilizing the membrane lipid composition. Salt stress often increases non-bilayer lipids (e.g., phosphatidylcholine), disrupting membrane integrity [33]. BR may counteract this by promoting the formation of bilayer-forming lipids like phosphatidylcholine, thereby reducing MDA accumulation [34,35]. Additionally, BR’s activation of glutathione metabolism and ion transporters could indirectly limit ROS generation by maintaining the ionic balance [36,37]. While these findings highlight BR’s potential in enhancing salt tolerance, the precise molecular interactions require further investigation. In conclusion, BR pretreatment correlated with reduced lipid peroxidation and an enhanced antioxidant capacity in rice seedlings under salt stress. These effects likely arise from BR’s dual role in activating antioxidant pathways and optimizing resource allocation for redox homeostasis. For the two different rice varieties, BR could regulate various pathways to maintain redox homeostasis, improve salt tolerance, and enhance the adaptation of rice to salt stress.

### 3.3. Exogenous Application of BR Improves Ionic Homeostasis in Rice Under Salt Stress

Under salt stress, excess Na^+^ in plants disrupts metabolic enzymes and induces the efflux of essential ions like K^+^ and Ca²^+^, leading to toxic protein accumulation [38]. Maintaining cytoplasmic K^+^ concentrations is critical for cellular function, as Na^+^/K^+^ ionic antagonism under stress reduces K^+^ uptake [39,40]. Ca²^+^ homeostasis also plays a pivotal role in salt acclimation, with K^+^/Na^+^ and Ca²^+^/Na^+^ ratios serving as key indicators of an ionic imbalance [41]. In this study, salt stress increased the leaf Na^+^ content and decreased K^+^/Ca²^+^ levels at the 4.5-leaf stage, with Chaoyouqianhao showing less pronounced changes than Huanghuazhan (Table 1). BR soaking under salt stress was associated with reduced Na^+^ levels and enhanced K^+^/Ca²^+^ contents in both varieties, improving the K^+^/Na^+^ and Ca²^+^/Na^+^ ratios. This aligns with BR’s reported association with Na^+^ efflux promotion and K^+^ channel regulation in guard cells [42].

Ionic homeostasis is closely linked to gene regulation. 9-Cis epoxycarotenoid dioxygenase (NCED) is a key regulator of abscisic acid (ABA) biosynthesis, with ABA promoting Na^+^ efflux and K^+^ uptake to maintain the intracellular ion balance [43]. In rice, five NCED genes (*OsNCED1*-*5*) exhibit distinct functions, with *OsNCED2* and *OsNCED3* critical for stress responses [44]. For example, *OsNCED3* overexpression reduced leaf Na^+^ accumulation while enhancing K^+^/Ca²^+^ retention under alkali stress, improving ion ratios [45]. Similarly, *OsNCED2* overexpression strengthened osmoregulation and the ion balance [46]. In this study, BR treatment upregulated *OsNCED2* expression in Huanghuazhan, suggesting a potential association with the BR-mediated ion balance through ABA signaling. The cell membrane contains various ion channels and transporter proteins that regulate ion influx and efflux, thereby maintaining the balance of ions both inside and outside the cell [39]. For instance, the rice high-affinity potassium transporter (*OsHKT1;5*), a Na^+^-selective transporter, maintains Na^+^/K^+^ homeostasis under salt stress by mediating Na^+^ efflux through xylem Na^+^ scavenging [47]. *OsHKT1;1*, another member of rice’s high-affinity K^+^ transporter family, is also associated with salt tolerance [48]. Under salt stress, the expression of *OsHKT2;1* was suppressed, reducing Na^+^ uptake and preventing its excessive accumulation [49]. In the current study, under salt stress, BR treatment downregulated the expression of *OsHKT2;1* and *OsHKT1;1* in Huanghuazhan, suggesting that BR alleviated the ionic toxicity by suppressing specific genes to limit Na^+^ accumulation in leaves.

### 3.4. The Exogenous Application of BR Improves Photosynthesis in Rice Under Salt Stress

Salt stress inhibits photosynthesis by altering enzyme activities, disrupting the chloroplast structure, limiting electron flow, and generating reactive oxygen species (ROS) [50]. Previous studies have shown that BR enhanced the photosynthetic pigment content and improved the photosynthetic capacity under low-temperature stress by increasing light energy capture efficiency in tungro [51], rice [52], and maize [53]. Our findings align with these reports; under salt stress, BR not only improved the SPAD values of both rice varieties but also induced the differential expression of 23 and 25 photosynthesis-related genes in Huanghuazhan and Chaoyouqianhao, respectively (Appendix A). Notably, genes encoding ferredoxin-reducing proteins (*OsFd5* and *OsFdC1*)—key components of the photosynthetic electron transport chain—were upregulated. Ferredoxins play a central role in shuttling electrons from photosystem I to enzymes like NADP^+^ reductase, balancing energy conversion and reducing power allocation [54]. In salt-stressed spinach, enhanced ferredoxin activity was associated with improved cyclic electron flow and reduced ROS accumulation, highlighting its dual role in photosynthesis and redox homeostasis [55], which aligns with BR’s role in sustaining electron transport under stress in our study. Genes encoding photosystem II (PSII) oxygen-evolving enhancer proteins (OsPsbR1, OsPsbR2, and OsPsbP) were also upregulated. These proteins stabilize the PSII complex, which is critical for water oxidation and oxygen release [56]. In wheat, drought induced PsbR homologs to maintain PSII efficiency [57], whereas under heat stress, alginate induced the upregulation of PsbP, enhanced the stability of photosystem II, and mitigated stress damage to wheat chloroplasts [58]. These are consistent with our findings from BR-influenced salt stress studies in rice. The upregulation of C3 Calvin cycle genes—including the RuBisCO small subunit (*OsRBCS1*) and malate dehydrogenase (*OsMDH4.1*)—suggested improved CO_2_ fixation. OsRBCS1 is essential for assembling active RuBisCO, the enzyme driving carbon assimilation [59]. *OsMDH4.1*, which is involved in redox balance, supported energy metabolism by regulating malate shuttling, a mechanism linked to stress resilience in rice [60].

Moreover, maintaining photosynthetic organ longevity is critical for stress tolerance. Studies in wheat and sorghum have linked “stay-green” traits to prolonged leaf function and enhanced yield under stress [61,62]. In maize, genes involved in cell wall biosynthesis and NAC transcription factors were identified as key regulators of this trait [63]. In this study, genes encoding alginic acid-6-phosphate synthase (*OsTPS1*, *OsTPS5*, *OsTPS6*, *OsTPS10*, and *OsTPS11*) and several NAC family genes (*OsNAC77*, *OsNAC095*, *OsNAC022*, *OsNAC103*, *ENAC1*, and *OsNAC111*) were significantly induced by BR under salt stress. OsTPS-mediated trehalose synthesis was linked to enhanced membrane stability under stress [64], while NAC genes integrated BR and ABA signaling to regulate stress-responsive pathways [65]. Additionally, the upregulation of *OsPORA*, a chlorophyll synthesis gene, suggested that BR might extend leaf photosynthetic activity by delaying chlorophyll degradation. In summary, BR treatment under salt stress influenced photosynthetic electron transport, key protein functions, CO_2_ fixation, and photosynthetic organs’ longevity, collectively enhancing rice leaves’ photosynthetic efficiency.

### 3.5. Roles of JA and Lipid Metabolism in Salt Stress

Jasmonic acid (JA) is a key lipid-derived hormone mediating plant salt stress responses [66]. This study identified four JA-related genes (*OsOPR1, OsJAZ2, OsJAZ11*, and *OsJAR2*) with altered expression under BS treatment. *OsOPR1* and *OsJAR2* participated in JA-Ile biosynthesis [67,68], while *OsJAZ2*/*OsJAZ11* regulated JA signaling via negative feedback, potentially enhancing antioxidant and stress-responsive pathways [69].

Our metabolomic data revealed a BR-induced JA metabolic divergence between varieties: JA levels decreased in salt-sensitive Huanghuazhan but increased in tolerant Chaoyouqianhao, suggesting genotype-specific tolerance strategies linked to JA metabolism (Table 5). This indicated that their distinct salt tolerance strategies might be associated with the differential JA metabolic patterns between the two rice varieties upon BR application, which aligns with previous reports on grapevine [70]. Xue et al. [71] demonstrated significant alterations in lipid metabolism and gene expression profiles in rice roots under salt stress, indicating that enhanced lipid metabolism might also contribute to improved salt tolerance. Furthermore, the GO analysis linked biofilm stability under BS treatment to rice salt tolerance. Integrated omics showed that Chaoyouqianhao (salt-tolerant rice) was enriched in lipid metabolism pathways, including cutin, suberine, and wax biosynthesis (ko00073); α-linolenic acid metabolism (ko00592); and fatty acid metabolism (Figure 7B,D), with upregulated genes like *OsACX2*, *OsLOX8*, and *OsClo5* (Appendix A). These findings suggested that salt-tolerant plants might better adapt to salt stress through lipid metabolism regulation, which was consistent with the results of Xu et al. [72]. Notably, BR treatment activated the JA biosynthesis module in the α-linolenic acid pathway in both varieties (Appendix A). In Chaoyouqianhao, the JA precursor 12-OPDA and its upstream gene *OsLOX8* were upregulated, reflecting the LOX-mediated oxygenation of α-linolenic acid to form 12-OPDA—an intermediate in JA-Ile synthesis [73,74]. The current findings demonstrated that the LOX-OPDA module critically regulated plant defense, JA biosynthesis, and stress adaptation. BR enhanced salt tolerance in rice by affecting JA signal transduction, metabolism, and lipid metabolism.

## 4. Materials and Methods

### 4.1. Plant Growth and Treatment

Two rice (*Oryza sativa* L.) genotypes differing in salinity tolerance, “Chaoyouqianhao” (salt-tolerant) and “Huanghuazhan” (salt-sensitive), were used in this study and were provided by the College of Coastal Agricultural Sciences of Guangdong Ocean University (Zhanjiang City, China). Brassinolide (BR) [molecular formula, C_28_H_48_O_6_; relative molecular weight, 480.68; chemical name, (22R,23R,24S)-2α,3α,22,23-tetrahydroxy-24-methyl-B-homo-7-oxa-5α-cholestan-6-one] was supplied by Zhengzhou Nongda Biochemical Co. (Zhengzhou, China). Rice seeds were sterilized with 0.5% NaClO for 20 min, then rinsed with distilled water 3–5 times. The sterilized rice seeds were randomly divided into two groups and soaked with distilled water and a 0.1 mg·L^−1^ brassinolide solution for 24 h. (BR solution preparation method: 0.011 g of brassinolide of 90% purity in white crystalline form was weighed, 150 µL of ethyl alcohol of 99.7% purity was added and stirred to dissolve it, followed by the addition of 100 µL of Tween 20 to aid in solubilization, and finally distilled water was added to dilute the solution to 0.1 mg·L^−1^.) The concentration of BR used to soak seeds was based on our laboratory studies of saline stress in rice [11]. The soaked seeds were germinated in an incubator in the dark at 30 °C for 24 h. After germination, the rice seeds with consistent germination (completely breaking through the seed coat) were selected and sown in plastic pots with the same amount of substrate and cultivated in a greenhouse (Guangdong Ocean University, Zhanjiang City, Guangdong Province, China). Seventy-five seeds were sown in each pot. The rice seedlings were cultivated with a day/night cycle of 12 h/12 h at 30/25 °C, respectively, with a relative humidity of 60% and a light intensity of 600 µmol m^−2^ s^−1^. The culture medium used in this experiment was a mixture of red soil and river sand. [The soil nutrient status: pH (4.85), alkaline hydrolyzable nitrogen (43.71 mg kg^−1^), available phosphorus (108.81 mg kg^−1^), available potassium (27.47 mg kg^−1^), and organic matter (10.61 g kg^−1^)]. The red soil was sieved and mixed with the washed river sand in a ratio of 3:1 (*v*/*v*), and then the mixture was put into plastic flowerpots with a size of 21 cm × 18 cm × 17 cm (3 kg per pot). All the plastic flowerpots were watered with 1 L of tap water and placed in a solar greenhouse for sowing. The rice seedlings were treated with NaCl when they grew to the 1.5-leaf stage. The NaCl concentration was chosen from previous studies of rice pot experiments [75,76]. Taking clear water as the control, 1 L of an NaCl solution with a concentration of 3 g ·L^−1^ was poured into the pot every 3 days, and the third watering amount after every 2 irrigations was 3 times the amount of water required, of which 2 times the liquid flowed out naturally, thus maintaining a stable salt concentration.

There were 4 treatments for each variety: CK—control, soaking with distilled water + watering with 0 g·L^−1^ NaCl, B—soaking with BR + watering with 0 g·L^−1^ NaCl, S—soaking with distilled water + watering with 3 g·L^−1^(51.33 mM) NaCl, BS—soaking with BR + watering with 3 g·L^−1^ (51.33 mM) NaCl. Each treatment was arranged in a random complete block design with 30 replicates. Samples were taken at the 2.5-leaf stage (14 days after sowing) and the 4.5-leaf stage (28 days after sowing) to determine the morphological and physiological indicators of rice plants.

### 4.2. Measurement of Rice Seedling Growth Indicators

Rice plant height was measured using a ruler, and the stem width was determined with Vernier calipers. The rice plants were separated into aboveground and below-ground parts at the rhizome. The surface of the samples was rinsed with distilled water to remove any deposited impurities, followed by three rinses with deionized water. The aboveground and belowground parts were initially heated at 110 °C for 30 min to deactivate enzymes and then dried at 75 °C until a constant weight was achieved. The dry weights of the aboveground and belowground parts were then accurately measured.

### 4.3. Determination of the Root Morphology of Rice Seedlings

The root systems of rice seedlings were scanned using an Epson V800 scanner, and root images were analyzed using the WinRHIZO Basic Root Analysis System software (Regent Instruments, Inc., Quebec, QC, Canada) to determine the total root length, total root surface area, total root volume, and mean root diameter.

### 4.4. Measurement of Oxidative Stress and Antioxidant Indices

At the 2.5- and 4.5-leaf stages, rice seedling leaves were collected for biochemical analyses.

The leaves (0.5 g) were ground in liquid nitrogen, and then 10 mL of a pre-cooled phosphate buffer solution (0.05 mmol·L^−1^ PBS, pH 7.8) was added. The homogenate was ground and centrifuged at 4 °C and 6000× *g* for 20 min. The activities of superoxide dismutase (SOD), catalase (CAT), peroxidase (POD), and ascorbate peroxidase (APX) in the supernatant were measured. Determination of antioxidant enzyme system: SOD activity was assessed using the NBT photochemical reduction method described by Spychalla [77]; CAT activities were determined by following the protocol of Ekinci et al. [78]; and the APX activity level was calculated according to the method described by Nakano [79].

The thiobarbituric acid (TBA) [80] assay was employed to quantify malondialdehyde (MDA) levels. We collected and weighed 0.5 g samples of seedling root and leaf tissues. These samples were ground in liquid nitrogen to form a powder. Then, the powder was mixed with 10 mL of phosphate buffer (pH 7.8) and extracted using 0.6% TBA prepared in 10% trichloroacetic acid (TCA). The mixture underwent heating in boiling water at 100 °C for 15 min before rapidly cooling on ice. Following centrifugation at 7000× *g* for 20 min, we measured the absorbance of the supernatant at wavelengths of 450 nm, 532 nm, and 600 nm.

The ascorbic acid (ASA) content was determined according to the method described by Arakawa [81], and the glutathione (GSH) content was measured by following the method of Nakano [79]. Absorbance values were measured at 534 nm and 412 nm, respectively.

### 4.5. Determination of the Ion Contents in the Leaves of Rice Seedlings

The leaves of rice seedlings were dried in an oven at 110 °C for 30 min and then at 75 °C until a constant weight was achieved. The dried samples were ground into a homogeneous powder using a tissue grinder. A 0.2 g aliquot of each sample was subjected to microwave digestion and an ashing pretreatment to prepare the test solution. The Na^+^, K^+^, and Ca²^+^ concentrations were determined using an inductively coupled plasma optical emission spectrometer (ICP-OES, Thermo Scientific ICAP 6000 Series Thermo Fisher Scientiffc, Waltham, MA, USA).

### 4.6. Determination of the SPAD Value of Rice Seedling Leaves

The SPAD (Soil and Plant Analyzer Development) value is an instrumental reading that measures the ratio of the amount of light reflected to the amount of light transmitted from a leaf using a chlorophyll meter. The chlorophyll content was measured using a SPAD chlorophyll meter (502 DL PLUS, Spectrum Technologies, Inc., Plainfield, IL, USA). The SPAD values were recorded from the most recently fully expanded functional leaves [82].

### 4.7. Determination of Chlorophyll Fluorescence Parameters

Chlorophyll fluorescence parameters, including the initial fluorescence (Fo), maximum fluorescence (Fm), maximum photochemical efficiency (Fv/Fm), and potential photochemical activity (Fv/Fo), were measured using a PAM-2500 portable chlorophyll fluorometer (Heinz Walz GmbH, Effeltrich, Germany).

### 4.8. Total RNA Isolation and Transcriptome Analysis

At the 1.5-leaf stage, 24 h after the completion of the salt stress treatment, inverted bifoliate leaves were collected from three groups: the clear water control group, the salt stress treatment group, and the combined BR and salt stress treatment group, with three biological replicates for each treatment. This study extracted RNA from rice leaves using a CTAB-PBIOZOL reagent followed by ethanol precipitation. The quality and quantity of total RNA were assessed using a Drop spectrophotometer and the Agilent 2100 Bioanalyzer (Thermo Fisher Scientific, Waltham, MA, USA) [83]. Clean reads were obtained using SOAPnuke [84], and a comparative analysis was performed using Bowtie2 against reference gene sequences. Differentially expressed genes (DEGs) with |Log2FC| ≥ 1 and *p* < 0.05 were identified using Phyper and subsequently analyzed through Gene Ontology (GO) and Kyoto Encyclopedia of Genes and Genomes (KEGG) enrichment analyses.

### 4.9. Real-Time Quantitative Fluorescence PCR (qRT-PCR) Validation

Real-time quantitative polymerase chain reaction (RT-qPCR) was employed to validate the expression levels of 11 randomly selected differentially expressed genes (DEGs) identified in the transcriptome using ACTIN as the internal reference gene. The sequences corresponding to the genes were obtained from the rice genome sequence database (TIGR), and the sequences of the exons of each gene were used to design the RT-PCR primers using Primer6.0 software (Appendix A). As described previously, the same RNA samples for the transcriptomic analysis were used for RT-PCR. The SYBR Green Premix Pro Taq HS qPCR Kit (Novizan Biotechnology Co., LTD, Nanjing, China) was used for the RT-qPCR assay using 20 mL reaction solutions containing 10 mL of 2 × SYBR real-time PCR premixture, 0.4 mL of primer F, 0.4 mL of primer R, 1 mL of cDNA and RNase free dH_2_O up to 20 mL. The qPCR reactions involved denaturation at 95 °C for 5 min followed by 40 cycles of 15 s at 95 °C and 30 s at 60 °C. The RT-qPCR assays were carried out using a QuantStudio Real-Time PCR system (Applied Biosystems, Foster City, CA, USA). The qPCR data were analyzed using the 2−DDCt quantitative method to determine differences in gene expression [85]. Three independent biological replicates and three technological replicates were used for each sample in this study.

### 4.10. Metabolite Extraction and Metabolomic Analysis

Seedlings were sampled from three treatment groups, the clear water control group, the salt stress treatment group, and the BR combined with the salt stress treatment group, at 24 h post-treatment, with three biological replicates per group. Freeze-dried samples were ground into a fine powder, and 50 mg aliquots were precisely weighed and transferred to Eppendorf tubes. Each aliquot was mixed with 700 μL of extraction solution (methanol/water = 3:1, pre-cooled to −40 °C). The samples were vortexed for 30 s, homogenized at 35 Hz for 4 min, and sonicated in an ice-water bath for 5 min. This homogenization and sonication process was repeated three times. Subsequently, the samples were incubated on a shaker at 4 °C overnight. After the incubation, the samples were centrifuged at 12,000 rpm (R = 8.6 cm) for 15 min at 4 °C. The supernatant was carefully filtered, and the resulting filtrate was diluted five-fold with a methanol/water mixture (*v*:*v* = 3:1), vortexed for 30 s, and transferred to 2 mL glass vials. A 30 μL aliquot from each sample was pooled to create a quality control (QC) sample. All samples were stored at −80 °C until the UHPLC-MS analysis. For this study, the untargeted metabolomic analysis was conducted using LC-MS/MS technology. Raw mass spectrometry data (raw files) were collected and imported into Compound Discoverer 3.1 (Thermo Fisher Scientific, USA) for processing. The Metabolome Information Analysis Process was utilized for data preprocessing, including the statistical analysis, metabolite classification, and functional annotation.

### 4.11. Statistical Analysis

Statistical analyses were performed using Excel 2016 and SPSS 19.0. One-way analysis of variance (ANOVA) followed by Duncan’s multiple comparison test were used to assess differences among treatments. The results are presented as means (X) ± standard errors (SEs). Graphs were generated using Origin 2019b software, and different lowercase letters indicate significant differences between treatments (*p* < 0.05).

## 5. Conclusions

The current study investigated the morpho-physiological, transcriptomic, and metabolomic changes underlying the specific mechanisms of the Chaoyouqianhao (salt-tolerant) and Huanghuazhan (salt-sensitive) varieties’ resistance to salt stress after the application of BR. The results indicated that BR universally alleviated salt stress in both varieties by regulating morphogenesis, enhancing the antioxidant capacity, maintaining ion balance, and improving photosynthetic efficiency (Figure 8).

However, the specific molecular mechanisms are somewhat different. In Huanghuazhan (salt-sensitive), BR primarily mitigated the ion imbalance by upregulating OsNCED2 to enhance ABA signaling and downregulating *OsHKT1;1/OsHKT2;1* to suppress Na^+^ accumulation while promoting K^+^ uptake. Antioxidant defenses were bolstered through the induced expression of CAT, POD, and APX, reducing ROS-induced damage. For photosynthesis, BR upregulated genes encoding photosystem II proteins (PsbR and PsbP) and Calvin cycle enzymes (RuBisCO and MDH), stabilizing electron transport and CO_2_ fixation. In Chaoyouqianhao (salt-tolerant), BR leveraged lipid metabolism and JA signaling for stress adaptation. Key pathways included activated α-linolenic acid metabolism, where upregulated *OsLOX8* drove 12-oxo-phytodienoic acid (12-OPDA) production—a JA precursor—via the LOX-OPDA module. This enhanced JA biosynthesis and defense responses. Additionally, BR upregulated genes involved in cutin/suberine biosynthesis (*OsACX2* and *OsClo5*), reinforcing cell membrane stability. While both varieties shared BR-induced antioxidant and photosynthetic enhancements, Chaoyouqianhao uniquely utilized lipid–JA crosstalk to strengthen stress resilience. These findings highlight that BR employed conserved stress mitigation strategies (e.g., antioxidant and photosynthetic regulation) across rice varieties but engaged distinct molecular pathways shaped by inherent salt tolerance traits. The current study will provide a framework and valuable empirical data for further studies on the relationship between the BR treatment and plant salt tolerance mechanisms.

## Figures and Tables

**Figure 1 plants-14-01555-f001:**
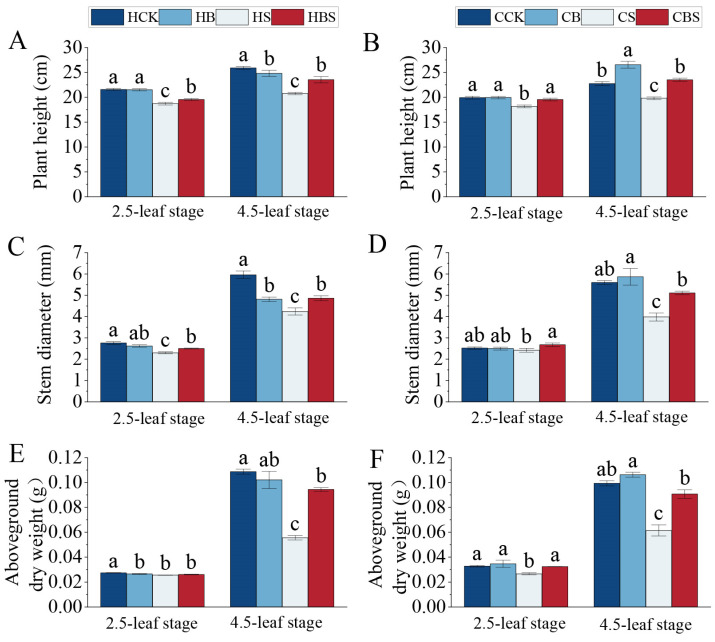
Effects of BR treatments on the phenotypes of the aboveground parts of Huanghuazhan and Chaoyouqianhao at the 2.5- and 4.5-leaf stages under salt stress. Plant heights of Huanghuazhan (**A**) and Chaoyouqianhao (**B**); stem base widths of Huanghuazhan (**C**) and Chaoyouqianhao (**D**); and dry weights of the aboveground parts of Huanghuazhan (**E**) and Chaoyouqianhao (**F**). H is the prefix for Huanghuazhan; C is the prefix for Chaoyouqianhao. CK: control, soaking with distilled water + watering with 0 g·L^−1^ NaCl. B: soaking with BR + watering with 0 g·L^−1^ NaCl. S: soaking with distilled water + watering with 3 g·L^−1^ NaCl. BS: soaking with BR + watering with 3 g·L^−1^ NaCl. The data are the means of three replicates ± SEMs. Different letters in the data columns indicate significant differences according to Duncan’s test (*p* < 0.05).

**Figure 2 plants-14-01555-f002:**
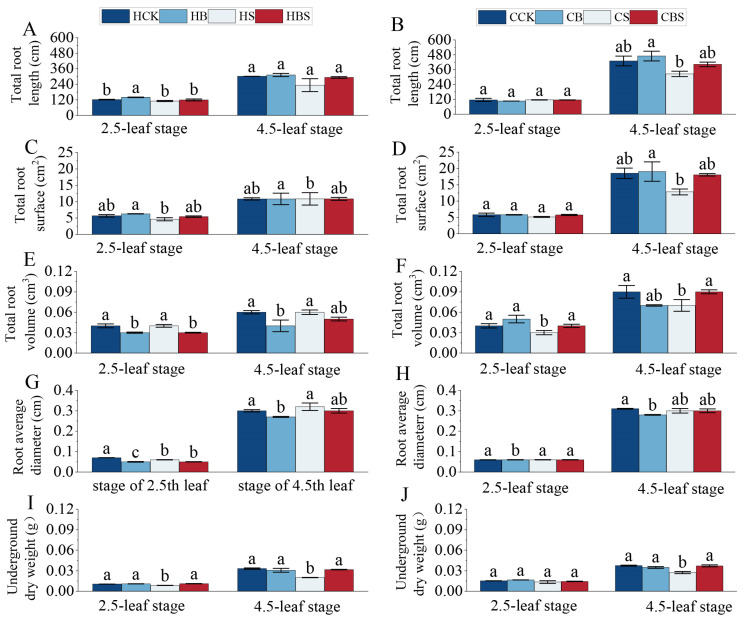
Effect of the BR treatment on the phenology of the underground parts of Huanghuazhan and Chaoyouqianhao at the 2.5- and 4.5-leaf stages under salt stress. Total root lengths of Huanghuazhan (**A**) and Chaoyouqianhao (**B**); total root surfaces of Huanghuazhan (**C**) and Chaoyouqianhao (**D**); total root volumes of Huanghuazhan (**E**) and Chaoyouqianhao (**F**); mean root diameters of Huanghuazhan (**G**) and Chaoyouqianhao (**H**); and the underground dry weights of Huanghuazhan (**I**) and Chaoyouqianhao (**J**). H is the prefix for Huanghuazhan; C is the prefix for Chaoyouqianhao. CK: control, soaking with distilled water + watering with 0 g·L^−1^ NaCl. B: soaking with BR + watering with 0 g·L^−1^ NaCl. S: soaking with distilled water + watering with 3 g·L^−1^ NaCl. BS: soaking with BR + watering with 3 g·L^−1^ NaCl. The data are the means of three replicates ± SEMs. Different letters in the data columns indicate significant differences (*p* < 0.05) according to Duncan’s test.

**Figure 3 plants-14-01555-f003:**
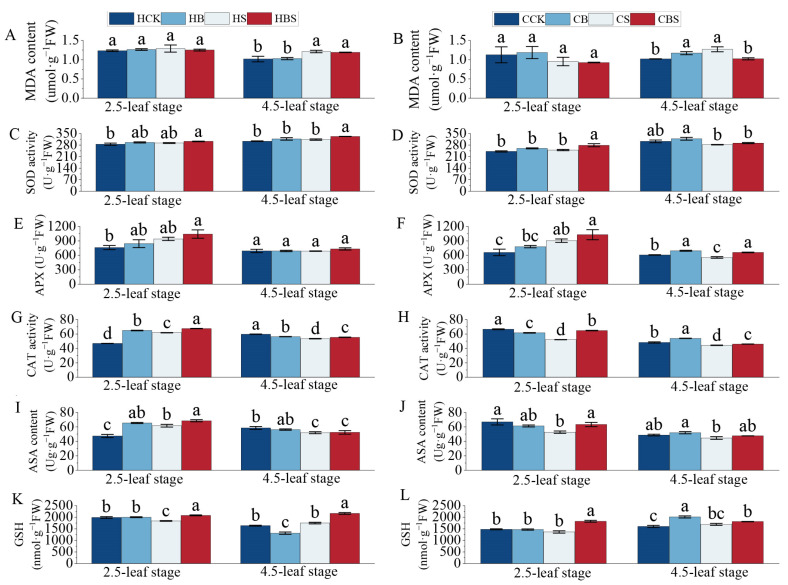
Effects of the BR treatment on membrane peroxidation and antioxidant system of leaves of Huanghuazhan and Chaoyouqianhao at the 2.5- and 4.5-leaf stages under salt stress. MDA contents in Huanghuazhan (**A**) and Chaoyouqianhao (**B**); SOD activities in Huanghuazhan (**C**) and Chaoyouqianhao (**D**); APX activities in Huanghuazhan (**E**) and Chaoyouqianhao (**F**); CAT activities in Huanghuazhan (**G**) and Chaoyouqianhao (**H**); ASA contents in Huanghuazhan (**I**) and Chaoyouqianhao (**J**); and GSH contents in Huanghuazhan (**K**) and Chaoyouqianhao (**L**). CK: control, soaking with distilled water + watering with 0 g·L^−1^ NaCl. B: soaking with BR + watering with 0 g·L^−1^ NaCl. S: soaking with distilled water + watering with 3 g·L^−1^ NaCl. BS: soaking with BR + watering with 3 g·L^−1^ NaCl. The data are the means of three replicates ± SEMs. Different letters in the data columns indicate significant differences (*p* < 0.05) according to Duncan’s test.

**Figure 4 plants-14-01555-f004:**
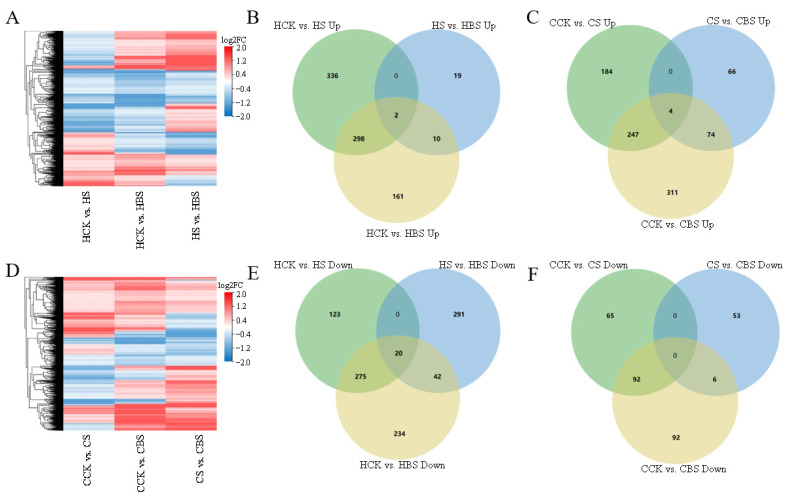
Clustering heat map and Venn diagram of differentially expressed genes in Huanghuazhan (**A**) and Chaoyouqianhao (**D**) under different treatments. Huanghuazhan seedlings: Venn diagram of upregulated genes (**B**) and downregulated genes (**E**) after various treatments. Chaoyouqianhao rice seedlings: Venn diagrams of upregulated (**C**) and downregulated (**F**) genes after various treatments. H is the prefix for Huanghuazhan; C is the prefix for Chaoyouqianhao. CK: control, soaking with distilled water + watering with 0 g·L^−1^ NaCl. B: soaking with BR + watering with 0 g·L^−1^ NaCl. S: soaking with distilled water + watering with 3 g·L^−1^ NaCl. BS: soaking with BR + watering with 3 g·L^−1^ NaCl. Differentially expressed genes were determined using Log2FC ≥ 1 and *p* < 0.05.

**Figure 5 plants-14-01555-f005:**
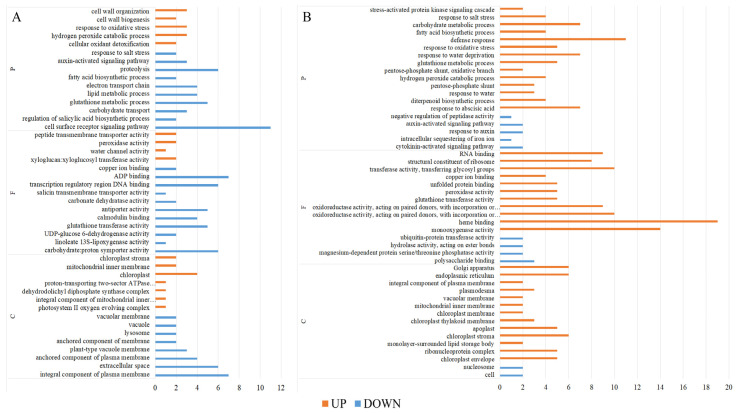
GO enrichment analysis of up- or downregulated genes in BR and salt-treated rice seedlings. Huanghuazhan: upregulated and downregulated unique GO enrichment classes (**A**). Chaoyouqianhao: upregulated and downregulated unique GO enrichment classes (**B**). The *x*-axis denotes the number of differentiated genes, and the y-axis denotes the GO enrichment annotation.

**Figure 6 plants-14-01555-f006:**
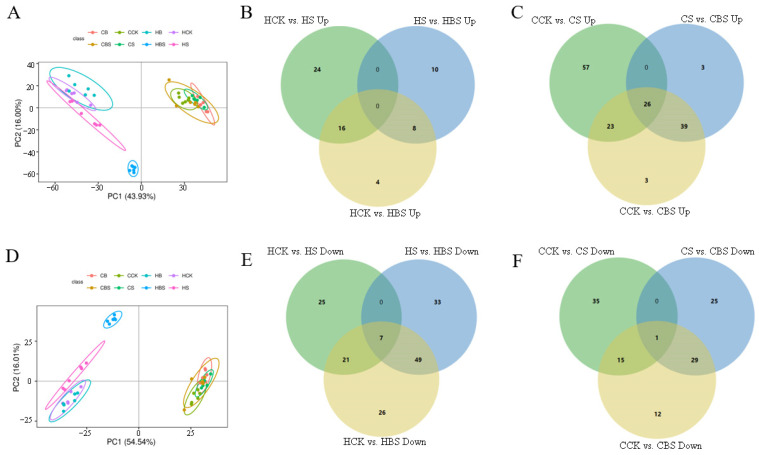
Principal component analysis (PCA) of differential metabolite expression after different treatments in Huanghuazhan and Chaoyouqianhao and Venn diagrams of differential metabolites in various groups. Positive ion mode PCA (**A**). Negative ion mode PCA (**D**). Huanghuazhan: Venn diagrams showing the increases and decreases in metabolite abundance after different treatments (**B**,**E**). Chaoyouqianhao: Venn diagrams showing the increases and decreases in metabolite abundance after different treatments (**C**,**F**). H is the prefix for Huanghuazhan; C is the prefix for Chaoyouqianhao. CK: control, soaking with distilled water + watering with 0 g·L^−1^ NaCl. B: soaking with BR + watering with 0 g·L^−1^ NaCl. S: soaking with distilled water + watering with 3 g·L^−1^ NaCl. BS: soaking with BR + watering with 3 g·L^−1^ NaCl. The differential metabolites were screened using the filters VIP ≥ 1, fold change ≥ 1.2 or ≤0.83, and a *p*-value < 0.05.

**Figure 7 plants-14-01555-f007:**
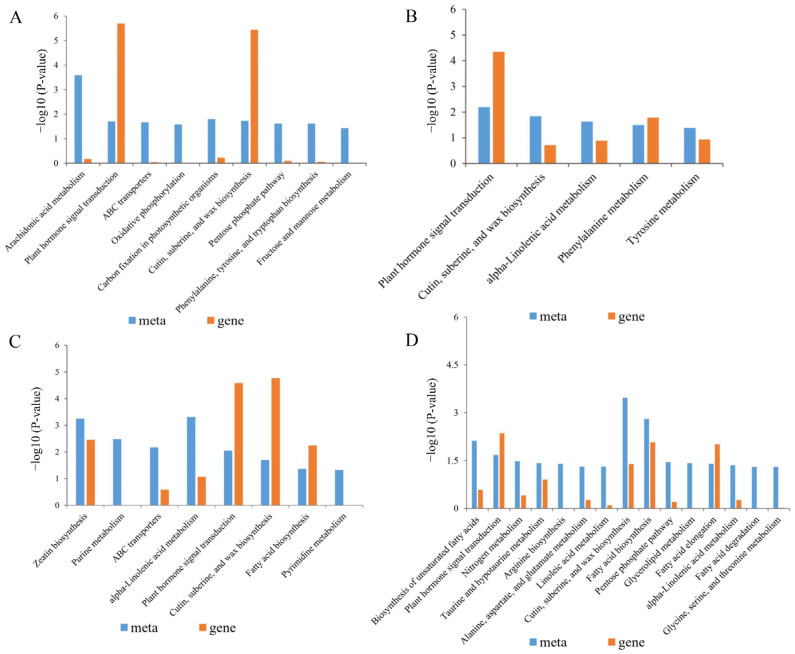
Differential expression of regulatory genes and metabolites mapped to KEGG pathways in the control vs. salt treatment groups and control vs. brassinolide + salt treatment groups. Huanghuazhan: (**A**) enriched pathways in the salt treatment and (**B**) the brassinolide + salt treatment group. Chaoyouqianhao: (**C**) enriched pathways in the salt treatment and (**D**) brassinolide + salt treatment groups.

**Figure 8 plants-14-01555-f008:**
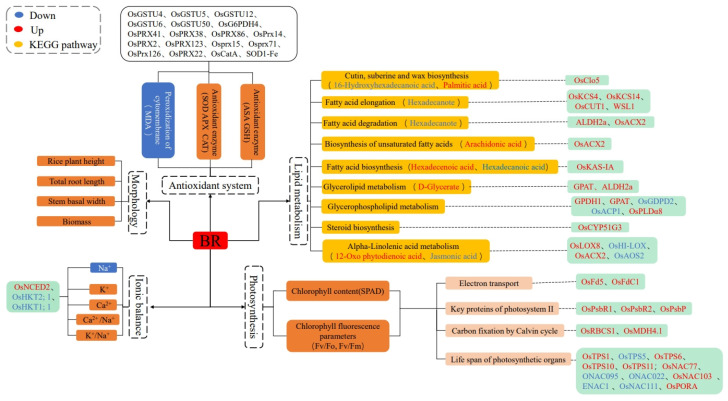
Mechanisms of BR for mitigating salt stress in rice.

**Table 1 plants-14-01555-t001:** Effects of BR on the Na^+^, K^+^, and Ca^2+^ content and K^+^/Na^+^ and Ca^2+^/Na^+^ of rice seedlings under salt stress.

Variety	Stage	Treatment	Na^+^ (mg·g^−1^)	K^+^ (mg·g^−1^)	Ca^2+^ (mg·g^−1^)	K^+^/Na^+^ (mg·g^−1^)	Ca^2+^/Na^+^ (mg·g^−1^)
Huang	2.5-leaf stage	CK	0.70 ± 0.08 b	2.48 ± 0.03 a	3.63 ± 0.03 a	3.64 ± 0.39 a	5.33 ± 0.52 a
huazhan		B	0.69 ± 0.01 b	1.97 ± 0.03 c	3.42 ± 0.01 b	2.87 ± 0.039 b	4.99 ± 0.01 a
		S	1.24 ± 0.02 a	1.81 ± 0.02 d	2.89 ± 0.02 d	1.46 ± 0.035 c	2.33 ± 0.05 b
		BS	1.16 ± 0.01 a	2.10 ± 0.01 b	3.05 ± 0.07 c	1.81 ± 0.01 c	2.63 ± 0.07 b
	4.5-leaf stage	CK	0.51 ± 0.00 c	1.89 ± 0.01 a	3.12 ± 0.01 b	3.71 ± 0.03 b	6.12 ± 0.01 b
		B	0.37 ± 0.00 c	1.71 ± 0.01 b	3.69 ± 0.01 a	4.59 ± 0.04 a	9.91 ± 0.03 a
		S	11.12 ± 0.11 a	1.21 ± 0.01 c	2.53 ± 0.02 d	0.11 ± 0.00 d	0.23 ± 0.01 d
		BS	5.43 ± 0.05 b	1.69 ± 0.00 b	3.02 ± 0.01 c	0.31 ± 0.00 c	0.55 ± 0.01 c
Chaoyou	2.5-leaf stage	CK	0.72 ± 0.04 c	2.41 ± 0.02 c	3.66 ± 0.03 a	3.35 ± 0.18 a	5.09 ± 0.23 a
qianhao		B	0.89 ± 0.03 b	2.48 ± 0.015 b	3.68 ± 0.01 a	2.79 ± 0.08 b	4.13 ± 0.14 b
		S	1.86 ± 0.02 a	1.66 ± 0.00 d	3.01 ± 0.03 b	0.89 ± 0.01 d	1.62 ± 0.01 c
		BS	1.83 ± 0.01 a	2.54 ± 0.02 a	3.00 ± 0.01 b	1.39 ± 0.02 c	1.64 ± 0.01 c
	4.5-leaf stage	CK	0.68 ± 0.03 c	2.13 ± 0.00 a	3.96 ± 0.01 b	4.07 ± 0.01 a	7.58 ± 0.01 b
		B	0.81 ± 0.03 c	2.01 ± 0.02 b	4.58 ± 0.07 a	3.41 ± 0.02 b	7.80 ± 0.02 a
		S	10.63 ± 0.07 a	1.64 ± 0.01 d	2.97 ± 0.01 d	0.16 ± 0.00 d	0.29 ± 0.01 d
		BS	5.25 ± 0.01 b	1.85 ± 0.02 c	3.27 ± 0.03 c	0.37 ± 0.01 c	0.64 ± 0.01 c

Different lowercase letters indicate significant differences at the *p* < 0.05 level. CK: control, soaking with distilled water + watering with 0 g·L^−1^ NaCl. B: soaking with BR + watering with 0 g·L^−1^ NaCl. S: soaking with distilled water + watering with 3 g·L^−1^ NaCl. BS: soaking with BR + watering with 3 g·L^−1^ NaCl.

**Table 2 plants-14-01555-t002:** Effects of BR on the leaf SPAD and chlorophyll fluorescence of rice seedlings under salt stress.

Variety	Stage	Treatment	SPAD	Fv/Fo	Fv/Fm
Huanghuazhan	2.5-leaf stage	CK	32.56 ± 0.27 a	1.17 ± 0.05 a	0.52 ± 0.01 a
		B	31.03 ± 0.40 b	1.16 ± 0.05 a	0.51 ± 0.01 a
		S	27.93 ± 0.48 c	0.46 ± 0.03 c	0.40 ± 0.02 a
		BS	31.36 ± 0.38 ab	0.84 ± 0.13 b	0.50 ± 0.08 a
	4.5-leaf stage	CK	22.31 ± 1.16 a	1.84 ± 0.01 b	0.65 ± 0.02 b
		B	23.99 ± 0.72 a	1.99 ± 0.03 a	0.74 ± 0.04 a
		S	18.09 ± 0.32 b	1.13 ± 0.01 c	0.57 ± 0.02 b
		BS	21.92 ± 0.23 a	1.49 ± 0.02 b	0.62 ± 0.01 b
Chaoyouqianhao	2.5-leaf stage	CK	32.43 ± 0.08 a	0.93 ± 0.03 b	0.48 ± 0.02 a
		B	30.96 ± 0.17 b	1.35 ± 0.03 a	0.47 ± 0.02 a
		S	27.60 ± 0.26 c	0.64 ± 0.02 c	0.46 ± 0.01 a
		BS	29.96 ± 0.17 ab	0.84 ± 0.05 b	0.52 ± 0.03 a
	4.5-leaf stage	CK	23.15 ± 0.57 a	0.62 ± 0.10 a	0.35 ± 0.04 a
		B	24.05 ± 0.78 a	0.62 ± 0.05 a	0.33 ± 0.01 a
		S	17.82 ± 0.37 b	0.32 ± 0.01 b	0.27 ± 0.02 a
		BS	22.59 ± 0.46 a	0.51 ± 0.05 ab	0.34 ± 0.04 a

The SPAD value reflects the relative amount of chlorophyll in the leaf; Fv/Fm is the maximum photochemical quantum yield of PSII; and Fv/Fo is the PSII potential photosynthetic efficiency. Different lowercase letters indicate significant differences at the *p* < 0.05 level. CK: control, soaking with distilled water + watering with 0 g·L^−1^ NaCl. B: soaking with BR + watering with 0 g·L^−1^ NaCl. S: soaking with distilled water + watering with 3 g·L^−1^ NaCl. BS: soaking with BR + watering with 3 g·L^−1^ NaCl.

**Table 3 plants-14-01555-t003:** The number of differentially expressed genes in rice seedlings.

Combinations	Upregulation	Downregulation	All DEGs
HCK vs. HS	636	418	1054
HCK vs. HBS	471	571	1042
HS vs. HBS	31	353	384
HB vs. HCK	80	27	107
CCK vs. CS	435	157	592
CCK vs. CBS	636	190	826
CS vs. CBS	114	59	203
CB vs. CCK	156	39	195

H is the prefix for Huanghuazhan; C is the prefix for Chaoyouqianhao. CK: control, soaking with distilled water + watering with 0 g·L^−1^ NaCl. B: soaking with BR + watering with 0 g·L^−1^ NaCl. S: soaking with distilled water + watering with 3 g·L^−1^ NaCl. BS: soaking with BR + watering with 3 g·L^−1^ NaCl. Differentially expressed genes were determined using Log2FC ≥ 1 and *p* < 0.05.

**Table 4 plants-14-01555-t004:** Breakdown of differential metabolites for both rice species.

Group	Total	Up	Down
HCK vs. HS	146	40	106
HCK vs. HBS	119	28	91
HS vs. HBS	80	18	68
CCK vs. CS	104	53	51
CCK vs. CBS	160	103	57
CS vs. CBS	144	89	55

H is the prefix for Huanghuazhan; C is the prefix for Chaoyouqianhao. CK: control, soaking with distilled water + watering with 0 g·L^−1^ NaCl. B: soaking with BR + watering with 0 g·L^−1^ NaCl. S: soaking with distilled water + watering with 3 g·L^−1^ NaCl. BS: soaking with BR + watering with 3 g·L^−1^ NaCl. The differential metabolites were screened using the filters VIP ≥ 1, fold change ≥ 1.2 or ≤0.83, and a *p*-value < 0.05.

**Table 5 plants-14-01555-t005:** Co-enriched pathways of the transcriptome and metabolome in rice seedlings treated with brassinolide + salt.

Varieties	Pathway	Metabolite ID	Metabolite Name	Log_2_FC
Huanghuazhan	Cutin, suberine, and wax biosynthesis (ko00073)	19.007_272.23482	16-Hydroxyhexadecanoic acid	−1.34
	Plant hormone signal transduction (ko04075)	13.729_210.12555	Jasmonic acid	−7.10
	α-Linolenic acid metabolism (ko00592)	13.729_210.12555	Jasmonic acid	−7.10
Chaoyouqianhao	Cutin, suberine, and wax biosynthesis (ko00073)	19.007_272.23482	16-Hydroxyhexadecanoic acid	−1.21
		21.193_256.24002	Palmitic acid	−0.51
	alpha-Linolenic acid metabolism (ko00592)	14.49_292.20360	12-Oxo phytodienoic acid	1.14
	Plant hormone signal transduction (ko04075)	10.37_210.12588	Jasmonic acid	0.68

## Data Availability

The transcriptome and metabolome data presented in this study can be found in online repositories. The raw sequence data have been deposited in the Genome Sequence Archive in the National Genomics Data Center, Beijing Institute of Genomics, Chinese Academy of Sciences, under accession number CRA023731, and are publicly accessible at https://ngdc.cncb.ac.cn/gsa (accessed on 15 May 2025). The metabolite data have been deposited in the National Genomics Data Center, Beijing Institute of Genomics, Chinese Academy of Sciences under accession number OMIX009472, and are publicly accessible at https://bigd.big.ac.cn/omix (accessed on 16 March 2025).

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
