# Peer review of "Integrated Analyses Reveal the Physiological and Molecular Mechanisms of Brassinolide in Modulating Salt Tolerance in Rice"

_plants, 2025, doi:10.3390/plants14101555_

Round 1
Reviewer 1 Report (Previous Reviewer 1)
Comments and Suggestions for Authors
The manuscript entitled “Integrated Analyses Reveal the Physiological and Molecular Mechanisms of Brassinolide in Modulating Salt Tolerance in Rice” presents an interesting aspect of the use of brassinolide in improving rice tolerance to salinity. Numerous analyses were carried out, including the physiological, metabolic and genetic levels, which demonstrated the beneficial effect of the hormone on the functioning of two rice varieties under salt stress conditions.
Generally, I did not find any serious shortcomings in the prepared manuscript. However, I have a few minor comments and some suggestions regarding the results and conclusions.
In my opinion, it is not explained anywhere (including the methodology) that the abbreviation CK means control. Similarly, has the abbreviation EBR been explained?
The description of the results has been prepared in such a way that it is not clear which of the given values, expressed in % of control, refer to one or another variety, and in some cases to the variety and developmental state. An example: “Compared to CK, the S treatment significantly decreased the contents of K+ (25%, 36% and 31%, 23%) and Ca2+(20%, 19% and 18%, 25%) at both periods of both rice varieties, respectively”. This needs correction.
I propose to move the explanation of the SPAD abbreviation to the methodology, leaving in the description of Table 2 only the information that this value reflects the chlorophyll content.
I do not see the point of describing the data in Table 4 (354-363) in detail. It is an exact repetition of the entire contents of the table. Moreover, in the next paragraph, some of these results are described again. In my opinion, it is worth emphasizing the comparison of salinity conditions with simultaneous treatment of plants with salt and brassinolide, which was missing.
It should be considered whether it would not be better to move Figures 8 and 9 to the supplement.
In the final version of the manuscript, the figures should be of better quality (clearer, larger fonts).
I propose that the Authors reconsider their conclusions. In my opinion, the beginning of the second paragraph repeats the information from the first paragraph, expanding it. It would be better to pay more attention to the differences between the varieties.
The text requires careful proofreading. In my opinion, some of the expressions are not correct. The example: “salt stress was significantly higher in salt-sensitive rice varieties…”, “jasmonic acid (JA) is an important class of lipid-derived hormones …”, “data analysis showed that BR-induced JA was down-regulated…”, or “JA signaling transduction”.
Comments on the Quality of English LanguageEnglish needs to be improved.
Author Response
Please see the attachment

Reviewer 2 Report (New Reviewer)
Comments and Suggestions for Authors
The manuscript “ Integrated Analyses Reveal the Physiological and Molecular Mechanisms of Brassinolide in Modulating Salt Tolerance in Rice” investigated the role of brassinolide (BR) in modulating salt tolerance mechanisms in rice (Oryza sativa). The study integrates physiological assessments with molecular analyses, including transcriptomics and metabolomics, to investigate how BR influences plant responses to saline stress. The research aims to deepen the understanding of hormonal regulation of stress adaptation in crops.
The article is well-structured, clearly divided into Introduction, Results, Discussion, and Methods. Logical flow between sections enhances readability.
Key words: The keywords must not be the same as in the title of the work. Please change:
brassinolide; rice. Additionaly, please organise the keywords alphabetically.
Results: The manuscript is very comprehensive and comprehensively examines the effects of salinity stress on rice varieties categorised as resistant and sensitive to the stress under study. The authors studied the morphological and physiological response of the plants and also performed analysis of key genes. The results are presented clearly and readily, the figures are well described.
Discussion: In the discussion, which is the most important chapter of the paper, the authors focused too much on describing their results, while too little attention was paid to explaining the mechanisms involved under the stress and attempting to neutralise it using BR. I suggest shortening the section in this paragraph on the description of own results. Please also better discuss the results obtained and compare them with more publications. The subject studied is well researched and there is a lot of current literature in this area.
Materials and Methods: Please add some detailed information about methods described in paragraph “Measurement of oxidative stress and antioxidant indices”.
Author Response
Please see the attachment.

Reviewer 3 Report (New Reviewer)
Comments and Suggestions for Authors
The following issues should be revised to improve the manuscript:
1. Language and writing style need to be improved. The manuscript suffers from numerous grammatical errors and awkward phrasing.
Line 36 Use italic font for Oryza sativa
2. The discussion contains overgeneralizations, such as linking all observed changes to BR without confirming causality. Therefore, authors should emphasize correlation rather than causation where appropriate.
3. Although RT-qPCR validation was mentioned, actual data were in supplementary materials and were not discussed meaningfully in the main text. Please add a summary of RT-qPCR results in the results section.
4. Why were 3 g·L⁻¹ NaCl and BR soaking selected for the study? Was this concentration pre-optimized? If it was, please add the citation for this.
5. Did BR affect gene expression under non-saline conditions? Some BR-only comparisons are underexplored.
6. Were any phenotypic trade-offs, such as growth vs. defense, recorded in BR-treated plants?
7. Unify the units and terminology
Line 577-578: Please add 2 more related references for this sentence.
Language and writing style need to be improved (major revision).
Round 2
Reviewer 2 Report (New Reviewer)
Comments and Suggestions for Authors
Thank you very much for addressing my comments and including them in the text of the publication.
This manuscript is a resubmission of an earlier submission. The following is a list of the peer review reports and author responses from that submission.
Round 1
Reviewer 1 Report
Comments and Suggestions for Authors
The manuscript entitled „Integrated Analyses Reveal the Physiological and Molecular Mechanisms of Brassinolide in Modulating Salt Tolerance in Rice” focuses on the role of brassinolide in rice adaptation to salinity. I found the topic interesting and the results obtained may be useful in agricultural practice. The manuscript contains a large number of results that include analyses at physiological, metabolic and genetic levels to provide a complete picture of the changes occurring in rice in response to brassinolide treatment under salt stress conditions. In my opinion, however, the results were not properly and clearly described and several aspects require significant improvement.
One class of phytohormones are brassinosteroids, and brassinolide is a member of this group of regulators (line 47). Generally, BR is the abbreviation for the entire group, while BL is used for brassinolide.
The Introduction refers to the work of Mu et al. 2022, which addresses a similar research topic (rice, salinity, BR). Therefore, I believe that when formulating the aim of the work, it was necessary to explain in detail what is the innovative nature of this research in relation to previous studies.
The results need to be re-described. The descriptions are inconsistent, some results are described in detail, others very generally, this should be unified. There were also various inaccuracies. Some of the results were described in such a way that it is difficult to determine where the mentioned percentages can be found in the charts/tables, e.g. Compared to the control (CK), salt treatment significantly decreased K+/Na+ and Ca2+/Na+ ratios by 60%, 97%, 73%, 96%, 56%, 96%, 68%, and 96%, respectively (lines 176-178) and many more similar examples. This is unacceptable. The authors use abbreviations that they do not explain, e.g. SPAD - what does it mean, why does it determine the chlorophyll level, what is the unit… The same description scheme for each studied aspect (figure) should be adopted, taking into account the BR effect on two cultivars under both control and stress conditions.
Are changes of a few % statistically significant? The authors performed a statistical analysis, but it is not always taken into account when interpreting the results.
The results of transcriptomic analyses are described too extensively. Are the data listed on lines 214-217 the sum obtained for both varieties? If so, what is their purpose and how do they relate to the data in Table 3? In my opinion, the fragment covering lines 252-286 is unnecessary. The titles of subsections 2.6 and 2.7 generally mean the same, I propose combining or changing them. What does the sentence: A total of 826 metabolites were identified across all samples (line 342) mean? What is the purpose of including the last two diagrams/figures if they are not described (they were commented on in one sentence, line 417).
The figures are too small and unreadable. Other abbreviations are used in figures and legends. More precision is required in this respect. There is an error in the numbering of the figures, 7 appears twice. Similarly, an error occurred in the legend of Fig. 3 (there are L and M instead of K and L). The figure number is missing in line 357. In addition to the table title, table legends should also include a short description.
When reporting, discussing and summarizing the results, insufficient emphasis was placed on the differences between two varieties differing in salt sensitivity. There was no reference to two developmental states, especially during the discussion - so why were both analysed? Some aspects of the discussion are repeated, e.g. the role of NCED. In my opinion, some of the results have been misinterpreted, this concerns e.g. the role of JA. JAZ and JAR proteins are not related to JA biosynthesis, but to JA signaling and metabolism. Moreover, why do the authors suggest that JA functions as a negative regulator in rice salinity tolerance (lines 608-610)?
The methodology lacks general information on: how long the plants were cultivated and which day they were subjected to 24-hour BR treatment (Plant growth and treatment). The Measurement of Physiological Indicators subsection does not concern physiological parameters, but oxidative stress/antioxidant levels. In my opinion, it is presented too briefly. Several times, unnecessarily, the following information appears at the beginning of the subsections of the methodology: Huanghuazhan and Chaoyouqianhao rice varieties were selected for the experiment. No information was provided regarding the qPCR reaction conditions.
The conclusions are rather a general summary and do not properly highlight the achievements of the work. The summary diagram is valuable, but does it show the differences between varieties? Apart from that, it was not mentioned anywhere.
The names of the journals are missing in the References.
The text contains various incorrect formulations/phrases, e.g. AOS does not mean allene oxide synthase but alginate oligosaccharides; the term treated with CK (control) cannot be used; the gene cannot catalyze the reaction and others. Abbreviations should be explained once, not every time they are used. Gene names should be written in italics. The accuracy of the English language is also questionable. For this reason, the entire manuscript needs to be carefully revised linguistically.
Comments on the Quality of English LanguageThe accuracy of the English language is also questionable.The entire manuscript needs to be carefully revised linguistically.
Author Response
Dear reviewer,
Thank you very much for your comments and professional advice. These opinions help to improve
the academic rigor of our article. Based on your suggestion and request, we have made correct
modifications to the revised manuscript. We hope that our work can be improved again. Furthermore,
we would like to show the details as follows.
Comment 1: One class of phytohormones are brassinosteroids, and brassinolide is a member of this
group of regulators (line 47). Generally, BR is the abbreviation for the entire group, while BL is
used for brassinolide.
Response 1: Thanks for your careful checks. We are sorry for our carelessness. Based on your
comments, we have made the corrections. Line48-50. (yellow marked)
Comment 2: The Introduction refers to the work of Mu et al. 2022, which addresses a similar
research topic (rice, salinity, BR). Therefore, I believe that when formulating the aim of the work,
it was necessary to explain in detail what is the innovative nature of this research in relation to
previous studies.
Response 2: Thank you for your suggestion. Valuable comments were agreed upon modified and
incorporated. We have added details to explain what is the innovative nature of this research to
previous studies. Line77-86. (yellow marked)
Comment 3: The results need to be re-described. The descriptions are inconsistent, some results are
described in detail, others very generally, this should be unified. There were also various
inaccuracies. Some of the results were described in such a way that it is difficult to determine where
the mentioned percentages can be found in the charts/tables, e.g. Compared to the control (CK), salt
treatment significantly decreased K/Na and Ca++ 2+/Na ratios by 60%, 97%, 73%, 96%, 56%, 96%,
68%, and 96%, respectively+
(lines 176-178) and many more similar examples. This is unacceptable.
The authors use abbreviations that they do not explain, e.g. SPAD - what does it mean, why does it
determine the chlorophyll level, what is the unit… The same description scheme for each studied
aspect (figure) should be adopted, taking into account the BR effect on two cultivars under both
control and stress conditions.
Response 3: Thanks for your careful checks. We are sorry for our carelessness in the result
description. Based on your comments, we have redescribed these results. The redescription of the
result section avoids the problem of inconsistencies and inaccuracies as far as possible, corrects the
percentage corresponding to the picture/table accurately, and supplements the problem of some
abbreviations that are not explained. Line89-243. (yellow marked)
Comment 4: Are changes of a few % statistically significant? The authors performed a statistical
analysis, but it is not always taken into account when interpreting the results.
Response 4: Thanks for your careful checks. We are sorry for our carelessness in interpreting the
results. In the process of redescribing the results section, we fully considered the statistical
significance in the interpretation of the results for percentage point changes in all outcomes. Line89-
243. (yellow marked)
Comment 5: The results of transcriptomic analyses are described too extensively. Are the data listed
on lines 214-217 the sum obtained for both varieties? If so, what is their purpose and how do they
relate to the data in Table 3? In my opinion, the fragment covering lines 252-286 is unnecessary.
The titles of subsections 2.6 and 2.7 generally mean the same, I propose combining or changing
them. What does the sentence: A total of 826 metabolites were identified across all samples (line
342) mean? What is the purpose of including the last two diagrams/figures if they are not described
(they were commented on in one sentence, line 417)?
Response 5: Thank you for your suggestion. Valuable comments were agreed upon modified and
incorporated.
(1) To address the problem that transcriptomics is too extensively described, we have carefully
revised and rephrased the results section of transcriptomics. Line261-349. (yellow marked)
(2) Regarding the l data description of “214-217”, we are sorry for the error in the description of the
number of differentially expressed genes. We have corrected this section in the process of
redescribing the transcriptomics results. Line261-268. (yellow marked)
(3) In response to “The titles of subsections 2.6 and 2.7 generally mean the same, I propose
combining or changing them”. We have revised this section accordingly. Line300-349. Figure 5.
(yellow marked)
(4) In response to “A total of 826 metabolites were identified across all samples”, the 826 differential
metabolites here are the differential metabolites identified in the two rice varieties in both positive
and negative ion modes, which were filtered to obtain the 753 differential metabolites used in the
analysis in our article, a section that we did not accurately represent in our previous manuscript, and
which we have now made changes to. Line369-272. (yellow marked)
(5) We add a description of Figures 8 and 9 and also analyze them later in the discussion section.
Line449-466. Line641-661. (yellow marked)
Comment 6: The figures are too small and unreadable. Other abbreviations are used in figures and
legends. More precision is required in this respect. There is an error in the numbering of the figures,
7 appears twice. Similarly, an error occurred in the legend of Fig. 3 (there are L and M instead of K
and L). The figure number is missing in line 357. In addition to the table title, table legends should
also include a short description.
Response 6: Thank you for your suggestion. Valuable comments were agreed upon modified and
incorporated.
(1) We have replaced all the images in the article where the numbers were too small and maximized
the numbers and text in the images. Figures 1-3 and Figure 7.
(2) We are sorry for the error in the numbering of the figures of the pictures in the text and have
made corrections in the appropriate places and added a short description of the table legend.
Line430,454,459,700,150;Line192-194; Line222-228; Line257-260; Line374-377; (yellow
marked)
Comment 7: When reporting, discussing and summarizing the results, insufficient emphasis was
placed on the differences between two varieties differing in salt sensitivity. There was no reference
to two developmental states, especially during the discussion - so why were both analysed? Some
aspects of the discussion are repeated, e.g. the role of NCED. In my opinion, some of the results
have been misinterpreted, this concerns e.g. the role of JA. JAZ and JAR proteins are not related to
JA biosynthesis, but to JA signaling and metabolism. Moreover, why do the authors suggest that JA
functions as a negative regulator in rice salinity tolerance (lines 608-610)?
Response 7: Thank you for your suggestion. Valuable comments were agreed upon modified and
incorporated.
(1) We are sorry for the lack of emphasis in the article on describing the differences between the
different salt-sensitive varieties. In the course of the extensive revision of the Results,
Discussion, and Conclusions sections, we focused on distinguishing the differences between
the two rice varieties.
(2) We have made changes to address the repetitive discussion of NCED: its discussion in
subsection 3.2 has been removed, and it is specifically discussed in subsection 3.3. Line559-
571. (yellow marked)
(3) Thank you very much for your reminder about the discussion of the role of JA, we have rediscussed subsection 3.5 and corrected the descriptions of this in the whole article; we found
that the view that “JA functions as a negative regulator in rice salinity tolerance” was inaccurate,
so we deleted this view and focused on the discussion of the role of JA in Chaoyouqianhao.
Line620-661. (yellow marked)
Comment 8: The methodology lacks general information on: how long the plants were cultivated
and which day they were subjected to 24-hour BR treatment (Plant growth and treatment).
The Measurement of Physiological Indicators subsection does not concern physiological
parameters, but oxidative stress/antioxidant levels. In my opinion, it is presented too briefly. Several
times, unnecessarily, the following information appears at the beginning of the subsections of the
methodology: Huanghuazhan and Chaoyouqianhao rice varieties were selected for the experiment.
No information was provided regarding the qPCR reaction conditions.
Response 8: Thank you for your carefulsuggestion. Valuable comments were agreed upon modified
and incorporated.
(1) In response to the lack of general information on Plant growth and treatment, we have
added details in the method section. Line663-696. (yellow marked)
(2) In response to “The Measurement of Physiological Indicators subsection does not concern
physiological parameters, but oxidative stress/antioxidant levels”, we have made a
correction. Line710. (yellow marked)
(3) We have removed this unnecessary information from this section. (Huanghuazhan and
Chaoyouqianhao rice varieties were selected for the experiment)
(4) We have added information regarding the qPCR reaction conditions, specific primer
designs and results in Supplementary Figure 2 and Table 3. Line747-763. (yellow marked)
Comment 9: The conclusions are rather a general summary and do not properly highlight the
achievements of the work. The summary diagram is valuable, but does it show the differences
between varieties? Apart from that, it was not mentioned anywhere.
Response 9: Thank you for your constructive advice on the conclusion section. We have rewritten
the results section and highlighted the achievements of the work and the differences between the
two breeds, and the re-edited conclusions section corresponds to the summary figure. (Figure 10)
Line791-820. (yellow marked)
Comment 10: The names of the journals are missing in the References.
Response 10: Thank you for your careful suggestion, we have added the journal names of all the
references. Line847-967
Comment 11: The text contains various incorrect formulations/phrases, e.g. AOS does not mean
allene oxide synthase but alginate oligosaccharides; the term treated with CK (control) cannot be
used; the gene cannot catalyze the reaction and others. Abbreviations should be explained once, not
every time they are used. Gene names should be written in italics. The accuracy of the English
language is also questionable. For this reason, the entire manuscript needs to be carefully revised
linguistically.
Response 11: Thank you for your careful suggestion, we are sorry for the various incorrect
formulations/phrases in the text. We have revised each of these incorrect formulas/phrases and have
made a carefully revised linguistically.
Comment 12: The accuracy of the English language is also questionable. The entire manuscript
needs to be carefully revised linguistically.
Response 12: Thank you for your suggestion. We have checked and corrected the English accuracy
of the entire text as much as possible, and the corrections are highlighted in yellow.
We acknowledge the need for major revisions and are committed to addressing the highlighted
issues comprehensively. We believe that these revisions will significantly strengthen our manuscript
and provide a more robust foundation for our claims.
We sincerely appreciate the time and effort invested by the reviewers in evaluating our manuscript!ee the attachment

Reviewer 2 Report
Comments and Suggestions for Authors
The article contains interesting and useful material, and as far as the content is concerned, I have no objection to its publication. However, it can only be published after serious revision:
- All the figures need to be redesigned: the text on them is too small and unreadable.
- P. 5, line 156: “13.08%, 23.91%, 33.67%, and 7.18%” should be replaced with “13%, 24%, 34%, and 7.2%”. The same in many other places.
- Table 1: Abbreviations HCK, HB, HS, HBS are somewhat confusing, it is better to replace them with CK, B, S, BS which are deciphered above.
- There are two Figures 7 in the article: one on page 13, the second on page 16.
- What does the text “(c) Kanehisa Laboratories” at the bottom of the pictures on pages 15 and 17 mean? These drawings are strikingly similar to the drawings:
https://t-stor.teagasc.ie/bitstream/handle/11019/243/1471-2164-13-193-S4.PDF?sequence=5&isAllowed=y
- The full chemical name of the compound referred to in the article as brassinolide (BR) should be given, as well as the company from which it was purchased.
- P. 21, line 612: “pathways[51]” should be replaced with “pathways [51]”. The same in many other places.
- P. 21, in chapter “Materials and Methods” authors should describe a method for the preparation of BR water solution. It is important because it’s known that brassinolide is insoluble in water.
- References: [J] appears after each article title, it should be removed and substituted by the name of journal.
Author Response
Dear reviewer,
Thank you very much for your comments and professional advice. These opinions help to improve
the academic rigor of our article. Based on your suggestion and request, we have made correct
modifications to the revised manuscript. We hope that our work can be improved again. Furthermore,
we would like to show the details as follows.
Comment 1: All the figures need to be redesigned: the text on them is too small and unreadable.
Response 1: Thank you for your suggestion. We have replaced all the images in the article where
the numbers were too small and maximized the numbers and text in the images. Figures 1-3 and
Figure 7.
Comment 2: P. 5, line 156: “13.08%, 23.91%, 33.67%, and 7.18%” should be replaced with “13%,
24%, 34%, and 7.2%”. The same in many other places.
Response 2: Thank you for your careful suggestion. We have revised this portion of the percentage
description and have revised it in other similar places in the manuscript. Many relevant points
throughout the text (yellow marked)
Comment 3: Table 1: Abbreviations HCK, HB, HS, HBS are somewhat confusing, it is better to
replace them with CK, B, S, BS which are deciphered above.
Response 3: Thank you for your careful suggestion. We have carefully revised the table to address
the confusion of treatment numbers by replacing HCK, HB, HS, HBS with CK, B, S, BS. Table1-2.
Line191-192 and 221-222 (yellow marked)
Comment 4: There are two Figures 7 in the article: one on page 13, the second on page 16.
Response 4: Thank you for your careful suggestion. We are sorry for this error in Figure numbers.
We have scrutinized the numbering of the images throughout the manuscript and have corrected the
incorrect numbering. Line430,454,459,799 (yellow marked)
Comment 5: What does the text “(c) Kanehisa Laboratories” at the bottom of the pictures on pages
15 and 17 mean? These drawings are strikingly similar to the drawings:
https://t-stor.teagasc.ie/bitstream/handle/11019/243/1471-2164-13-193-
S4.PDF?sequence=5&isAllowed=y
Response 5: Thank you for your suggestion. The text “(c) Kanehisa Laboratories” Kanehisa
Laboratories is the copyright holder of the KEGG pathway map. The KEGG Pathway maps in our
manuscript were downloaded from the KEGG website, KEGG BRITE: KEGG Pathway Maps (https://www.kegg.jp/brite/query=01230&htext=br08901.keg&option=-a&node_proc=br08901_org&proc_enabled=map&panel=collapse) . We
labeled the downloaded images according to our own study and explained the labeling information
with a simple explain below the images. Figure 8 and Figure 9. Line 454-457, 459-462 (yellow
marked)
Comment 6: The full chemical name of the compound referred to in the article as brassinolide (BR)
should be given, as well as the company from which it was purchased.
Response 6: Thank you for your careful suggestion. We are sorry for the omission of this section
and have added it to the manuscript. Line 667-668 (yellow marked)
Comment 7: P. 21, line 612: “pathways[51]” should be replaced with “pathways [51]”. The same
in many other places.
Response 7: Thank you for your careful suggestion. We are sorry for this careless error and have
double-checked the entire text to correct it.
Comment 8: P. 21, in chapter “Materials and Methods” authors should describe a method for the
preparation of BR water solution. It is important because it’s known that brassinolide is insoluble in
water.
Response 8: Thank you for your careful suggestion. We have added the method for preparing an
aqueous solution of BR. Line 671-672 (yellow marked)
Comment 9: References: [J] appears after each article title, it should be removed and substituted
by the name of journal.
Response 9: Thank you for your careful suggestion. we have added the journal names of all the
references and removed [J]. Line847-968
We acknowledge the need for major revisions and are committed to addressing the highlighted
issues comprehensively. We believe that these revisions will significantly strengthen our manuscript
and provide a more robust foundation for our claims.
We sincerely appreciate the time and effort invested by the reviewers in evaluating our manuscript!

Round 2
Reviewer 1 Report
Comments and Suggestions for Authors
The authors responded to all comments included in the reviews. The manuscript was significantly improved. Despite this, I still have critical comments about some of the changes introduced and the language used.
The manuscript was not edited very carefully (commas, spaces, etc.). The correctness of the English wording and various oversights still require improvement. These are some examples: “when treated with BS treatment”, “rice seedlings seedlings”, “differential metabolites”, “the expression levels of the 11 transcriptome differential genes identified”, “BR treatment significantly reduced jasmonic acid abundance in Huanghuazhan but was significantly up-regulated in Chaoyouqi” and others.
Similarly, despite the changes introduced, I still find the following sentences unclear (which result refers to the variety or developmental state?):
“Salt treatment significantly reduced plant height, stem base width, and dry weight of above-ground parts compared with the control by 10-19%, 8-29%, and 7-49% in two rice varieties, at the 2.5th stage and 4.5th stage respectively (Figures 1A-F)” - lines 106-108
or
“Under salt stress, the total root surface and underground dry weight of Huanghuazhan and the total root volume of Chaoyouqianhao were significantly lower than CK at two periods, reduced by 19-21%, 19-39%, and 22-25% respectively (Figures 2C-J)” - lines 129-131
or
“When treated with BS treatment, compared to the S, the GSH content of two rice varieties increased significantly by 13.08%, 23.91%, and 33.67%, 7.18%, at the two stages respectively” - lines 174-176
Regarding 753 differential metabolites - how does this value relate to the data in table 4 (I still don't understand...).
In my opinion, it would be better to present the values of 2080% or 1463% as a multiple of the control values.
The conclusions have been improved. However, I think it would be much clearer to first present the similarities and then the differences between the two varieties.
What is the difference between the following conclusions:
“BR alleviates ion imbalance caused by salt stress in Huanghuazhan by upregulating an NCED-related gene and downregulating sodium accumulation-associated genes, thereby suppressing Na⁺ accumulation and promoting K⁺ uptake” - lines 803-80
and
“BR upregulates an NCED-related gene and downregulates sodium accumulation-associated genes to suppress Na⁺ accumulation and promote K⁺ uptake, thus alleviating ion imbalance caused by salt stress in Huanghuazhan” - lines 805-807.
In my opinion, the figures are still not very clear (fonts are too small). However, perhaps this issue does not raise any doubts for the Editor.
Comments on the Quality of English LanguageEnglish should be checked and improved.
Reviewer 2 Report
Comments and Suggestions for Authors
The authors were not careful enough in correcting the noted deficiencies in the manuscript. Although some of them are not very important (e.g. Comment 2, Rep. 1), some others are of critical value for the manuscript quality. Firstly, neither full chemical name of the compound referred to in the article as brassinolide (BR) nor its structural formule is given by the authors (Comment 6, Rep. 1). This easily could mean that the manuscript has nothing in common with brassinolide itself, and its title and content should be corrected. The same is true in respect to the Comment 8. Authors' response to it is unsatisfactory: what does small amount mean, what is No. 8 additive and what kind of alcohol in what amount was used??? The answer to this question allows characterizing the quality and reproducibility of the experiment performed.
Round 3
Reviewer 2 Report
Comments and Suggestions for Authors
I regret to note the authors' resolute unwillingness to answer my direct question about what substance they used in the work under the name brassinolide. The answer should be either the full chemical name of the compound used, or its structural formula, or better yet, both together, since the given authors' answers demonstrate their insufficient familiarity with the subject under study. The formula given in the latest version of the article corresponds to at least two natural brassinosteroids, and only one of them has the right to be called as brassinolide. Both are highly active phytohormones and are promising for practical use, but differ in availability and cost. The lack of an answer to the question about the chemical structure of the brassinosteroid studied in the work makes it impossible to discuss the extensive material of the article and calls into question its title and content. Thus, I recommend to reject the article.